# SEEPHYS: Does Seeing Help Thinking? – Benchmarking Vision-Based Physics Reasoning

**Kun Xiang**[1,§,*], **Heng Li**[1*], **Terry Jingchen Zhang**[2,Φ,*], **Yinya Huang**[2,5*],
**Zirong Liu**[1], **Peixin Qu**[1], **Jixi He**[1], **Jiaqi Chen**[4], **Yu-Jie Yuan**[3], **Jianhua Han**[3],
**Hang Xu**[3], **Hanhui Li**[1†], **Mrinmaya Sachan**[2,5], **Xiaodan Liang**[1†]

[§]Lead CS Team    [Φ]Lead Physics Team

[1]Sun Yat-sen University [2]ETH Zurich
[3]Huawei Noah's Ark Lab [4]The University of Hong Kong
[5]ETH AI Center

## Abstract

We present SEEPHYS, a large-scale multimodal reasoning benchmark grounded in physics spanning across middle school to PhD candidacy exams. The benchmark covers 7 fundamental physics domains and incorporates 21 categories of highly heterogeneous diagrams. In contrast to prior works where visual elements are often helpful but not necessary, our benchmark features a substantial proportion of vision-essential problems (75%) that mandate visual information extraction for problem-solving. Even best-performing reasoning models (e.g., Gemini-2.5-Pro) achieve sub-60% accuracy. These results reveal fundamental challenges in current MLLMs, particularly in: (i) establishing meaningful connections between diagram interpretation and physics reasoning, and (ii) overcoming their persistent reliance on textual cues as cognitive shortcuts.

**Project Page:** https://seephys.github.io/

**Hugging Face:** https://huggingface.co/datasets/SeePhys/SeePhys

## 1  Introduction

While mathematical reasoning has been a cornerstone for evaluating the reasoning capability of large language models (LLMs) and multimodal large language models (MLLMs) [8, 14, 11, 18, 21, 6, 23, 5, 50, 42, 45], the realm of natural science, especially the discipline of physics, remains understudied as a even more diverse testbed for complex scientific reasoning. Physics reasoning inherently binds text explanations to real-world visual contexts, exposing critical gaps in their ability to emulate human-like world modeling in the context of physics problem-solving [10, 24, 51].

Frontier models have begun to demonstrate abstract perception of physical scenarios and logical reasoning capabilities [25, 12, 29, 30, 9]. Due to the intrinsic diversity of physics diagrams and the fact that they inherently reflect the law of physics in our world, developing a comprehensive benchmark for evaluating physics reasoning abilities and cross-modal understanding is crucial for enhancing LLMs.

---

[*]These authors contributed equally to this work.

[†]Corresponding authors. Email: lihh77@mail.sysu.edu.cn, xdliang328@gmail.com

39th Conference on Neural Information Processing Systems (NeurIPS 2025) Track on Datasets and Benchmarks.

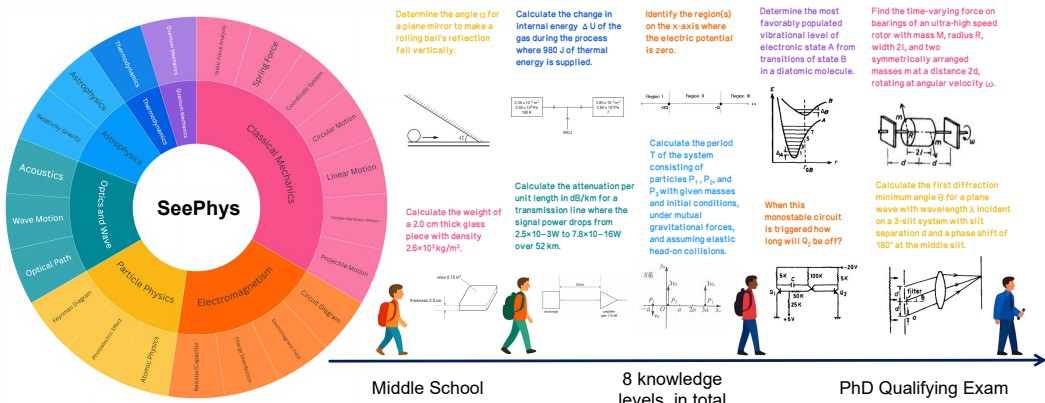

Figure 1: Overview of **SEEPHYS**. It encompasses 7 core physics domains and 21 diagram types, spanning the full knowledge spectrum from middle school to PhD candidacy exams levels.

Early research primarily focused on assessing physical commonsense [4] and basic scientific knowledge [22], which was later gradually extended to competition-level and university-level physics problems [34, 10]. Due to the broad knowledge scope and high annotation difficulty inherent, some studies only considered test samples from limited knowledge levels [52, 13, 48]. Furthermore, other works targeted the evaluation of language models and did not incorporate visual information [33, 10, 46]. Notably, compared to the mathematics domain, there has been limited exploration of models' perceptual differences in processing physics diagrams, despite their richer topological structures.

With these challenges in mind, we introduce **SEEPHYS**, to measure LLMs' capability to visually understand the law of physics. It is a fully multimodal benchmark for evaluating physics reasoning across middle school to PhD qualifying exam levels. **SEEPHYS** comprises 2,000 rigorously validated questions collected from open-source textbooks, exercises, examinations, and competitions. These questions span 7 major fields of both classical and modern physics. To assess the extent to which different models rely on visual information for reasoning, we curate two subsets with different degrees of visual information enrichment and additionally compile supplementary copies of 2,000 purely visual instances where all problem statements in texts are presented in picture form. Through meticulous selection of 21 diagram types by domain experts, each problem challenges frontier MLLMs to integrate domain knowledge with visual understanding of physics diagrams (e.g., Feynman diagrams for particle interactions and Circuit diagrams for Electromagnetism). With **SEEPHYS**, we conduct extensive experiments to evaluate 28 leading LLMs and MLLMs such as o4-mini [30] and Gemini-2.5-Pro [9]. The results reveal that even with extensive chain-of-thought, none of the current models could surpass 55% accuracy. Our analysis reveals that even non-essential diagrams can enhance physics reasoning performance when presented to MLLMs. We have also raised a challenge[3] in the 2nd AI for Math Workshop at ICML 2025[4].

Our main contributions are summarized as follows:

- We propose a purely multimodal benchmark spanning multiple knowledge levels and domains. The meticulously curated benchmark comprises 2,000 rigorously annotated questions paired with 2,245 images.

- By prioritizing physics' unique blend of observation and theoretical deduction, we assess how well models emulate the way human scientists observe, deduce, and understand complex natural phenomena. Our findings reveal significant gap in models' capabilities to leverage multimodal information.

- We conduct a comprehensive evaluation of current LLMs and MLLMs, followed by an in-depth analysis of their failure modes. Results demonstrate that even frontier models struggle with physics problems across different knowledge levels in various visual contexts.

---

[3] https://www.codabench.org/competitions/7925/
[4] https://sites.google.com/view/ai4mathworkshopicml2025/

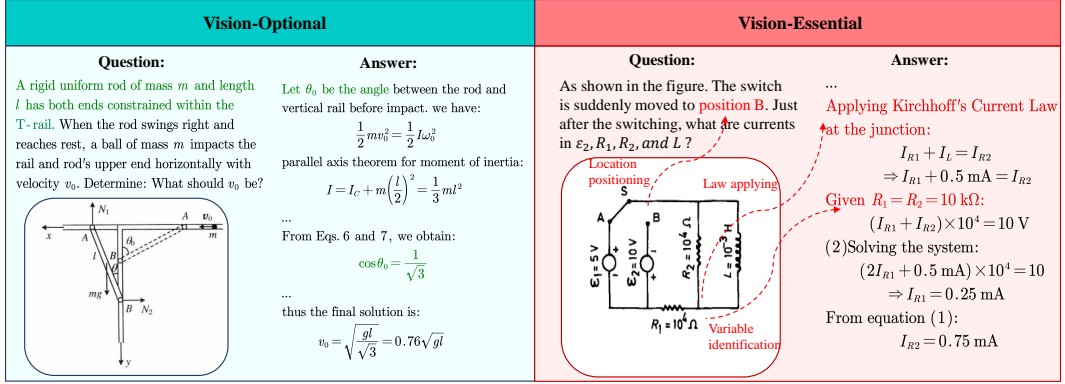

Figure 2: Examples of Vision-Optional/Vision-Essential questions. In Vision-Optional samples, texts provide sufficient visual descriptions (e.g., graphical attributes and spatial relationships) to help respondents with illustration. In Essential samples, images contain indispensable problem-solving information, such as numerical values for key variables and unspecified topological structures.

## 2 Related Work

**Math reasoning benchmarks.** Mathematical reasoning has emerged as a central testbed for evaluating LLMs. Early benchmarks like GSM8K [8] established the foundation for multi-step textual reasoning through elementary problems, while MATH [14] advanced the field with competition-level tasks (e.g., AMC/AIME), exposing critical limitations of early models. As these benchmarks achieved saturation, the community shifted toward higher complexity, e.g., introducing Olympiad-level challenges requiring formal theorem proving and combinatorial reasoning [11, 18]. Concurrently, the rise of multimodal reasoning spurred benchmarks such as MathVista [21] and MATH-V [42] to integrate visual comprehension (e.g., diagrams, plots) with compositional reasoning. However, MathVerse [50] has found that MLLMs tend to rely on the reasoning capabilities of language models when performing mathematical tasks. In contrast, scientific diagrams, which are abstractions derived from real-world scenarios, with their complex visual features, may provide a more effective testbed for benchmarking models' multimodal capabilities.

**Physics Reasoning benchmarks.** Contemporary physics benchmarks target broad scope of domains, with frontier datasets now encompassing: (1) PhD candidacy exam problems, and (2) Olympiad-style challenges. Text-based physics benchmarks like PHYBench [33] and UGPhysics [46] provide challenging problems that test advanced reasoning skills, yet fundamentally lack the visual components necessary to assess diagram interpretation abilities. Multimodal physics benchmarks such as PhysReason [52], OlympiadBench [13], and PHYSICS [10] emphasize visual reasoning challenges without analysis regarding the extent of visual components' impact. Moreover, constrained by the high annotation costs stemming from the extensive domain knowledge required and the scarcity of qualified multimodal materials, these datasets lack comprehensive coverage across knowledge hierarchies and detailed annotations of diagram types. To address these limitations, we contribute a dataset with fully multimodal and full-spectrum physics problems.

## 3 SEEPHYS

### 3.1 Data Collection Principles

The **SEEPHYS** benchmark aims to challenge current MLLMs from a multimodal perspective, especially visual understanding and reasoning capabilities of physics diagrams. The data collection adheres to the following principles:

**Visual information as a must.** We observe that the diagrams in existing data sources can be paired with the questions in two ways: (1) diagrams that substantially dominate the reasoning and problem-solving process of the problem (**Vision-Essential, VE**); (2) diagrams that are act as a supplement to help with illustration, but visual information does not play the major role in the thinking process (**Vision-Optional, VO**).

Table 1: Comparison of **SEEPHYS** and related datasets in physics. Mid: Middle school; High: High school; Olympiad: Beginner/advanced Olympiad; UG: Undergraduate/senior undergraduate; MA: Master's; PhD: PhD qualifying exams.

| | Images | Size | Mid | High | Olympiad | UG | MA | PhD |
|---|---|---|---|---|---|---|---|---|
| *Physics Benchmarks* | | | | | | | | |
| UGPhysics [46] | 0 | 11,040 | ✗ | ✗ | ✗ | ✓ | ✗ | ✗ |
| PHYSICS [10]* | 298 | 1,297 | ✗ | ✗ | ✗ | ✓ | ✓ | ✗ |
| PHYBench [33] | 0 | 500 | ✓ | ✓ | ✓ | ✓ | ✗ | ✗ |
| PhysReason [52] | 972 | 1,200 | ✗ | ✗ | ✓ | ✓ | ✓ | ✗ |
| *Physics Subset in Cross-domain Benchmarks* | | | | | | | | |
| ScienceQA [22] | 2,797 | 3,215 | ✓ | ✓ | ✗ | ✗ | ✗ | ✗ |
| OlympiadBench [13] | 1,958 | 2,334 | ✗ | ✓ | ✓ | ✗ | ✗ | ✗ |
| SciBench [44]* | 177 | 291 | ✗ | ✗ | ✗ | ✓ | ✗ | ✗ |
| SciEval [37] | 0 | 1,657 | ✓ | ✓ | ✗ | ✓ | ✗ | ✗ |
| MMMU [48] | 444 | 443 | ✗ | ✗ | ✗ | ✓ | ✗ | ✗ |
| MMMU-Pro [49] | 65 | 60 | ✗ | ✗ | ✗ | ✓ | ✗ | ✗ |
| GPQA [34] | 0 | 227 | ✗ | ✗ | ✗ | ✗ | ✗ | ✓ |
| ARB [35] | 31 | 129 | ✗ | ✗ | ✗ | ✗ | ✓ | ✓ |
| HLE [32] | 28 | 230 | ✗ | ✗ | ✗ | ✗ | ✓ | ✓ |
| **SEEPHYS** (Ours) | **2,245** | **2,000** | ✓ | ✓ | ✓ | ✓ | ✓ | ✓ |

\* Not fully open-source.

To analyze the performance variations of MLLMs across different types of visual information, the **SEEPHYS** benchmark implements rigorous categorization to distinguish them. We resort to experts in physics to annotate a problem as **Vision-Essential** with all essential information contained in graphical AND textual data, and to annotate a problem as **Vision-Optional** with all key information for problem-solving fully covered in text, as examples shown in Figure 2.

**Wide knowledge spectrum.** To provide a comprehensive evaluation of the model's physics knowledge comprehension and modeling capabilities, questions are systematically selected across the following eight progressive knowledge levels: middle school, high school, beginner Olympiad problems, advanced Olympiad problems, undergraduate, senior undergraduate, master, and PhD candidacy exams. Domain experts in physics are invited to selectively curate the most representative problems at each knowledge tier.

**Open-ended format without ambiguity.** The data format in **SEEPHYS** is set to open-ended questions, each with one single definitive answer, which reduces random guessing raised in the multiple-choice question setting and thereby obtaining more accurate scores. Therefore, during data collection, those problems with ambiguous expressions and multiple explainable solutions were filtered out.

Appendix C shows the detailed instructions for the annotators.

## 3.2 Annotation

**Collection and pre-processing.** We first collect more than 7000 PDF documents from publicly available textbooks, exercise problems, examinations, and competitions, as well as the International Physics Olympiad and Cambridge A-Level Physics. Data sources are from international educational systems, including Eastern Asia, Europe, North America, and many others. The resulting questions are diverse and multilingual (Chinese, English).

We then utilize Mathpix[5] to perform OCR parsing on the collected documents and convert them into markdown text. Afterwards, we construct the (`question`, `reasoning`, `answer`) triples for each question. GPT-4.1 [27] is employed to process redundant line breaks, string omissions, and LaTeX syntax errors. Finally, the trained annotators use a LaTeX parser to conduct manual verification on the triples.

---

[5] https://mathpix.com/

**Standardization.** Some of the source questions contain multiple independent sub-questions, which do not align with most of the questions in our dataset. Therefore, we manually segment the sub-questions into distinct components and recombine them with shared question stems. For those multiple-choice source questions, all are converted into the **SEEPHYS** standard open-ended response formats. For some numerical computation problems with decimal points in answers (e.g., 10.1), we provide the corresponding significant figures (e.g., =3). This mitigates potential approximation-induced errors. To prevent conceptual errors and ambiguity, each question instance is cross-validated by two independent annotators.

**Fine-grained categorization.** First, questions are coarsely classified into **7 subjects** based on their domain, as directly provided by the data source. To further analyze LLMs' sensitivity to different visual features, we introduce a fine-grained classification of **21 diagram types**. Through consultation with national curriculum standards and internal discussions with physics experts, we stratify all questions into **8 levels** based on knowledge spectrum. Table in Appendix A.1 lists the subjects, diagram types, difficulty levels, and detailed statistics. Notably, Olympiad competition problems exhibit significant variance in difficulty—we subdivide them into beginner/advanced Olympiad based on problem-solving durations. For undergraduate-level questions, we distinguish between standard undergraduate and senior undergraduate tiers according to whether they depend on mathematical physics methods. We then categorize the collected problems into **Vision-Essential** (75%) and **Vision-Optional** (25%) according to their levels of visual information enrichment. Graphical components in VE problems contain analytically indispensable information, e.g., circuit structure, kinematic diagrams with labeled vectors, or scale-dependent data visualizations. In contrast, diagrams in VO cases provide supplementary but non-critical information, e.g., a sketch of a physics scenario.

**Data leakage prevention.** To minimize the risk of data leakage, we eliminate samples with inconsistent responses by toggling the search function of GPT-4o [28] on and off. Inspired by Rein et al. [34], we subsequently conduct a manual search for the remaining questions with correct responses.

**Multimodal enhancement.** To further eliminate the influence of textual modality information, we first use o4-mini [30] to add a caption field to each example, containing a detailed description of geometric features and numerical information. We then augment the original 2,000 questions to obtain a purely visual version. Specifically, we render each question along with its corresponding diagrams into a single image up to 4,096×4,096 pixels. During rendering, we introduce variations in font types and sizes while ensuring readability based on the diagram's dimensions. The introduction of purely visual QA not only enhances the authenticity of evaluation but also advances the model's cross-modal understanding capabilities—mirroring human cognition by extracting key features from pixels, associating abstract concepts, and ultimately achieving problem-solving accuracy comparable to that under text-image QA conditions. Appendix A demonstrates the cases.

### 3.3 Data Analysis

**SEEPHYS** comprises 2,000 distinct questions paired with 2,985 diagrams (averaging 1.49 images per question). These questions comprehensively span 8 knowledge levels, 21 types of diagram categories, and 7 key physics fields. The detailed statistics are shown in Appendix A.

## 4 Experiments

### 4.1 Evaluation Protocol

To guide the models in generating reasoning-augmented responses, we design zero-shot Chain-of-Thought prompts among English and Chinese (Appendix D). To evaluate model performance across varying levels of visual information availability, we deploy four experimental settings:

- **Text+Vision, TV**: A question with the paired diagrams, as our baseline setting. The results reflect the model's ability to simultaneously *comprehend visual elements* and *process textual information*.

- **Text+Caption, TC**: A question with a diagram caption. The results reflect the model's capability to *process textual information* and *reconstruct graphical information from text*.

- **Text Only, TO**: Only a question text is given. The results represent the model's *pure text processing* capability without any visual input.

- **Vision Only, VO**: A composite image rendered from question text and the paired diagrams. The results reflect the model's ability to *interpret diagram elements* and *extract visual form text*.

Among them, VO setting directly uses 2,000 purely visual instances, and the remaining three are based on 2,000 multimodal instances. We conduct experiments on both Vision-Essential/Vision-Optional subsets across all four settings.

**Composite judgment strategy.** Recent advancements in the reasoning capabilities of LLMs enable them to discern key information within complex responses, thereby mitigating the inaccuracies often introduced by applying template matching to open-ended questions. Therefore, we further develop a composite judgment strategy based on a combination of LLM and template matching. As a first step, the model generates a response to the given input question and significant figures. Subsequently, the final answer is extracted through template matching and LLM-based processing. During the scoring process, SymPy[6] is first utilized to perform an initial screening for straightforward answers. Responses that do not pass the screening are then compared with the ground-truth using LLM. We apply accuracy as the metric for this deterministic evaluation. In the experiments in this paper, we use DeepSeek-V3 [16] as the extraction and judge model.

## 4.2 Evaluation Models

We conduct experiments with 8 NVIDIA A800 GPUs. To comprehensively evaluate the difficulty of **SEEPHYS** and the visual comprehension and reasoning capabilities of current AI systems, we employ a series of mainstream closed- and open-source models as baselines, including:

**9 Large language models:** DeepSeek-R1 [12], DeepSeek-V3 [16], Qwen3-235B-A22B [40], Qwen2.5-72B-Instruct [47], QwQ-32B [41], R1-Distilled-Llama-70B [12], Llama-4-Scout-17B [26], Gemma3-27B [38], Llama-3.1-8B [20]. We evaluate them with the TC setting.

**19 Multimodal large language models:** OpenAI o4-mini [30], OpenAI o3-mini [31], OpenAI o1 [29], Gemini-2.5-Pro [9], Claude 3.7 Sonnet [2], Doubao-1.5-pro [36], GPT-4.1 [27], GPT-4o [28], QvQ-72B-preview [39], Qwen-VL series [3, 43], Llama-3.2-Vision series [19], LLaVA-NeXT-7B [17], Phi-4-multimodal [1], InternVL2.5-8B [7], LLaVA-OneVision-7B [15]. All these MLLMs are benchmarked with the TV/TC/TO/VO settings.

We also conduct a systematic human evaluations on a stratified 200-question subset of SeePhys. For middle school to undergraduate-level items, we employ 3 physics major students who complete independent parallel trials, with their average score serving as the performance metric. For Master/PhD-level questions, we recruit 4 PhD candidates specializing in astrophysics, condensed matter physics, particle physics, and quantum optics respectively, computing the union of their correct answers as an indicator of optimal expert performance.

Detailed introduction and implementations of each model are in Appendix D.

## 4.3 Performance across Differential Knowledge Levels

**Multimodal physics reasoning is challenging.** Results in Table 2 demonstrate that **SEEPHYS** poses significant challenges to current mainstream models. Even state-of-the-art reasoning MLLMs (Gemini-2.5-Pro and o4-mini) achieve only under 55% accuracy, while other models, such as Doubao-1.5-pro and Claude-3.7-Sonnet, attain merely 43.9% and 34.6% respectively. These results clearly indicate substantial room for improvement in the physics reasoning capabilities of popular models. A surprising finding is that current LLMs demonstrate competitive performance, e.g., DeepSeek-R1 achieves an accuracy of 42.2%, which is comparable to the performance of o3-mini with multimodal inputs (40.3%). It suggests that the multimodal alignment capability of current MLLMs still has significant room for improvement.

**Diminishing returns of knowledge injection.** Table 2 further illustrates performance disparities across models at varying knowledge levels. Contrary to expectations, the difficulty progression for models does not strictly follow the knowledge level (e.g., senior undergraduate and advanced Olympiad questions exhibit lower accuracy than PhD candidacy exams). This discrepancy suggests that current models primarily rely on knowledge memorization rather than truly learning the derivation

---

[6]https://www.sympy.org

Table 2: Accuracy (%) of different LLMs/MLLMs by knowledge level. Mid: Middle school; High: High school; BO: Beginner Olympiad; AO: Advanced Olympiad; UG: Undergraduate; SUG: Senior undergraduate; MA: Master's; PhD: PhD qualifying exams. The highest and second-highest scores in each section are **bolded** and underscored, respectively. The performance of human experts (also bolded) achieved the highest accuracy of 86.5%, significantly outperforming the current best MLLM.

| Models | Mid | High | BO | AO | UG | SUG | MA | PhD | Total |
|---|---|---|---|---|---|---|---|---|---|
| *Large Language Models* | | | | | | | | | |
| Human Expert | **100.0** | **94.4** | **92.3** | **71.7** | **92.9** | **94.7** | **100.0** | **83.0** | **86.5** |
| DeepSeek-R1 [12] | **54.9** | **46.9** | **47.7** | **31.9** | **49.9** | **34.2** | **49.0** | **41.2** | **42.2** |
| DeepSeek-V3 [16] | 53.9 | 42.6 | 36.4 | 22.8 | 45.4 | 29.7 | 35.9 | 37.5 | 36.0 |
| R1-Distilled-Llama-70B [12] | 48.0 | 41.4 | 34.6 | 14.2 | 31.5 | 16.0 | 28.9 | 25.9 | 26.9 |
| Qwen3-235B-A22B [40] | 47.1 | 33.7 | 31.8 | 20.4 | 41.2 | 25.1 | 31.7 | 30.7 | 31.1 |
| Qwen2.5-72B-Inst [47] | 41.2 | 40.2 | 25.2 | 8.2 | 26.8 | 12.8 | 18.6 | 17.8 | 21.1 |
| QwQ-32B [41] | 47.1 | 42.2 | 44.9 | 15.5 | 40.0 | 20.1 | 32.4 | 24.0 | 29.7 |
| Llama-4-Scout-17B [26] | 48.0 | 36.5 | 31.8 | 11.3 | 28.5 | 14.2 | 28.3 | 26.1 | 24.8 |
| Gemma3-27B [38] | 21.6 | 36.5 | 30.8 | 5.1 | 23.1 | 9.1 | 15.2 | 11.9 | 16.9 |
| Llama-3.1-8B [20] | 26.5 | 15.7 | 17.8 | 3.9 | 7.6 | 3.7 | 10.3 | 8.4 | 9.2 |
| *Multimodal Large Language Models* | | | | | | | | | |
| Human Expert | **100.0** | **94.4** | **92.3** | **71.7** | **92.9** | **94.7** | **100.0** | **83.0** | **86.5** |
| o4-mini [30] | 66.7 | 61.8 | 56.1 | 41.8 | 53.8 | 45.7 | 51.0 | **53.4** | 51.9 |
| o3-mini [31] | 47.1 | 46.2 | 39.3 | 28.3 | 47.0 | 36.1 | 48.3 | 42.3 | 40.3 |
| o1 [29] | 60.8 | 56.6 | 50.5 | 32.5 | 54.4 | 40.6 | 52.4 | 40.4 | 45.6 |
| Gemini-2.5-Pro [9] | 69.6 | **66.7** | **64.5** | **46.7** | **64.2** | 50.2 | **53.8** | 44.2 | **54.9** |
| Claude-3.7-Sonnet [2] | 52.9 | 51.8 | 43.0 | 16.7 | 41.4 | 26.5 | 33.8 | 32.4 | 34.6 |
| Doubao-1.5-pro [36] | **70.6** | 58.2 | 49.5 | 29.2 | 56.6 | 34.7 | 40.7 | 37.5 | 43.9 |
| GPT-4.1 [27] | 51.0 | 52.6 | 41.1 | 17.0 | 39.7 | 31.1 | 42.1 | 35.6 | 35.3 |
| GPT-4o [28] | 37.3 | 39.0 | 34.6 | 7.5 | 23.4 | 15.5 | 24.1 | 21.8 | 21.9 |
| QVQ-72b-preview [39] | 38.2 | 36.5 | 30.8 | 11.3 | 25.9 | 14.2 | 26.2 | 20.2 | 22.5 |
| Qwen2.5-VL-72B-Inst [3] | 61.8 | 42.2 | 29.0 | 10.4 | 29.9 | 14.6 | 18.6 | 19.4 | 24.2 |
| Qwen2.5-VL-7B-Inst [3] | 39.2 | 25.3 | 21.5 | 4.2 | 8.7 | 5.9 | 10.3 | 7.3 | 11.6 |
| Qwen2.5-VL-3B-Inst [3] | 30.4 | 21.3 | 13.1 | 2.9 | 10.4 | 7.3 | 6.2 | 6.2 | 9.8 |
| Qwen2-VL-7B-Inst [43] | 24.5 | 17.3 | 14.0 | 4.4 | 8.5 | 4.6 | 10.3 | 7.0 | 9.2 |
| Llama-3.2-90B-Vision [19] | 21.6 | 25.7 | 22.4 | 3.9 | 9.3 | 10.0 | 12.4 | 8.9 | 11.7 |
| Llama3.2-11B-Vision [19] | 23.5 | 18.5 | 14.0 | 4.2 | 5.4 | 3.7 | 4.8 | 7.5 | 8.3 |
| LLaVA-NeXT-7B [17] | 14.5 | 12.7 | 11.2 | 5.5 | 13.2 | 8.2 | 11.0 | 9.4 | 8.7 |
| Phi-4-multimodal [1] | 20.6 | 12.4 | 12.1 | 4.4 | 7.0 | 5.0 | 8.3 | 4.9 | 7.6 |
| InternVL2.5-8B [7] | 17.6 | 12.4 | 9.3 | 2.9 | 5.6 | 3.2 | 4.1 | 5.1 | 6.2 |
| LLaVA-OneVision-7B [15] | 20.6 | 10.8 | 12.1 | 2.7 | 5.4 | 2.3 | 6.2 | 5.4 | 6.1 |

patterns of scientific laws. Furthermore, by examining results from middle school to PhD, we observe that weaker models demonstrate a substantially steeper performance reduction ratio (e.g., LLaVA-OneVision-7B, -74.5%) compared to stronger ones (e.g., o4-mini, -19.9%). Not only indicates that cutting-edge models fail to grasp fundamental principles underlying even simple physics concepts, but it also shows that knowledge injection has reached diminishing marginal returns.

## 4.4 Performance across Differential Visual Dependency Problems

**Does seeing help thinking? Impact of visual information on MLLM reasoning.** Our experiment in Table 3 employs four distinct settings to evaluate model performance under varying vision availability. Firstly, all the models in the vision-essential subset demonstrate a dependence on visual information, as evidenced by the high values of $\Delta_1$ and $\Delta_2$. Interestingly, even in the vision-optional subset where the images do not contain necessary information for solving problems, most models still exhibit performance improvements ($\Delta_2$=29.5% in o3-mini and 56.1% in Claude-3.7-Sonnet). This may be because physics diagrams, even when their intrinsic structures can be inferred from given text, can assist models in understanding abstract concepts and modeling real-world scenarios. This fundamentally distinguishes SEEPHYS data from structurally simple mathematical geometric figures.

Table 3: Accuracy (%) of different MLLMs under varying levels of visual information enrichment (with relative performance gaps). TV: Text+Vision. TC: Text+Caption. TO: Text Only. VO: Vision Only. $\Delta_1$: (TV-TC)/TV. $\Delta_2$: (TV-TO)/TV. $\Delta_3$: (TV-VO)/TV. The highest and second-highest scores in each section are **bolded** and underscored, respectively. The highest and lowest $\Delta$ are highlighted in green and red, respectively.

| Models | TV | TC | TO | VO | Avg | $\Delta_1$ | $\Delta_2$ | $\Delta_3$ |
|---|---|---|---|---|---|---|---|---|
| *Vision-Essential subset (75%)* | | | | | | | | |
| o4-mini [30] | 46.5 | **40.5** | 29.9 | **35.7** | **38.2** | 12.9 | 35.7 | 23.2 |
| o3-mini [31] | 32.9 | 15.3 | 16.3 | 7.1 | 17.9 | 53.5 | 50.4 | 78.4 |
| o1 [29] | 38.5 | 32.0 | 23.7 | 23.9 | 29.5 | 16.9 | 38.4 | 37.9 |
| Gemini-2.5-Pro [9] | **49.0** | 40.3 | **32.0** | 21.0 | 35.6 | 17.8 | 34.7 | 57.1 |
| Claude-3.7-Sonnet [2] | 27.9 | 22.8 | 12.3 | 20.2 | 20.8 | 18.3 | 55.9 | 27.6 |
| Doubao-1.5-pro [36] | 39.0 | 30.1 | 24.1 | 23.9 | 29.3 | 22.8 | 38.2 | 38.7 |
| GPT-4.1 [27] | 29.2 | 26.5 | 18.5 | 20.4 | 23.7 | 9.2 | 36.6 | 30.1 |
| GPT-4o [28] | 17.1 | 15.9 | 10.1 | 12.7 | 14.0 | 7.0 | 40.9 | 25.7 |
| QvQ-72B-preview [39] | 16.5 | 15.3 | 11.6 | 15.3 | 14.7 | 7.3 | 29.7 | 7.3 |
| Qwen2.5-VL-72B [3] | 18.0 | 16.4 | 12.1 | 9.3 | 13.0 | 8.9 | 32.8 | 48.3 |
| Qwen2.5-VL-7B [3] | 9.6 | 7.7 | 5.7 | 5.2 | 7.1 | 19.8 | 40.6 | 45.8 |
| Qwen2.5-VL-3B [3] | 6.8 | 6.8 | 6.9 | 3.3 | 7.1 | 0.0 | -1.5 | 51.5 |
| *Vision-Optional subset (25%)* | | | | | | | | |
| o4-mini [30] | 68.4 | **68.0** | 66.7 | **58.4** | **65.4** | 0.6 | 2.5 | 14.6 |
| o3-mini [31] | 62.4 | 33.6 | 44.0 | 12.8 | 38.2 | 46.2 | 29.5 | 79.5 |
| o1 [29] | 67.0 | 64.2 | 60.8 | 49.8 | 60.5 | 4.2 | 9.3 | 25.7 |
| Gemini-2.5-Pro [9] | **72.4** | 64.4 | **68.6** | 39.6 | 61.3 | 11.1 | 5.2 | 45.3 |
| Claude-3.7-Sonnet [2] | 53.8 | 47.6 | 23.6 | 41.6 | 41.7 | 11.5 | 56.1 | 22.7 |
| Doubao-1.5-pro [36] | 68.8 | 64.8 | 63.2 | 41.4 | 59.6 | 5.8 | 8.1 | 39.8 |
| GPT-4.1 [27] | 53.6 | 54.0 | 54.2 | 28.2 | 47.5 | -0.7 | -1.1 | 47.4 |
| GPT-4o [28] | 36.4 | 40.0 | 35.4 | 24.0 | 34.0 | -9.9 | 2.7 | 34.1 |
| QvQ-72B-preview [39] | 40.6 | 38.2 | 37.1 | 38.2 | 38.5 | 5.9 | 8.6 | 5.9 |
| Qwen2.5-VL-72B [3] | 42.8 | 40.0 | 38.6 | 22.0 | 31.3 | 6.5 | 9.8 | 48.6 |
| Qwen2.5-VL-7B [3] | 17.4 | 16.2 | 16.2 | 9.8 | 14.9 | 6.9 | 6.9 | 43.7 |
| Qwen2.5-VL-3B [3] | 18.8 | 15.2 | 14.2 | 8.4 | 14.2 | 19.1 | 24.5 | 55.3 |

**To what extent do models utilize visual perception?** Since the four settings assess the model's ability to leverage vision and text (TV), pure text reasoning (TC and TO), and vision-text recognition (VO), we further analyze the degree of visual dependency across different models in the vision-essential subset. With the Vision Only setting, o4-mini demonstrates high accuracy, while QvQ-72B-preview shows a relatively less reduction after removing textual information ($\Delta_3 = 7.3\%$), indicating that both graphical understanding and visual-text recognition contribute significantly to their reasoning. Furthermore, both o3-mini (78.4%) and Gemini-2.5-Pro (57.1%) exhibit very high $\Delta_3$, meaning that when utilizing visual information, they heavily rely on textual information recognition (poor OCR ability). On the other hand, we observe that some models exhibit more pronounced performance improvements when captions replace images (GPT-4o/4.1), suggesting they process textual information more efficiently—they rely more heavily on language model capabilities. In summary, different models exhibit distinct dependencies on graphical topology, visual-text recognition, and textual information understanding, which likely stems from differences in their multimodal training emphases.

**Impact of diagram types on model performance.** In Figure 3, we compare the performance of a powerful reasoning model, o4-mini, with a weaker open-source MLLM (Qwen2.5-VL-3B-Instruct) across different types of physics diagrams. With Text+Vision as baseline setting, even after excluding the maximum and minimum values, the accuracy range of o4-mini across various images remains widely dispersed (31.1%), indicating that the model may have specific effects on certain visual features. The significant gaps compared to the TO setting on *Wave Motion*, *Circuit Diagram*, and *Coordinate System* demonstrate that these diagram types specifically challenge models' multimodal reasoning capabilities. Furthermore, different models exhibit distinct strengths in processing specific

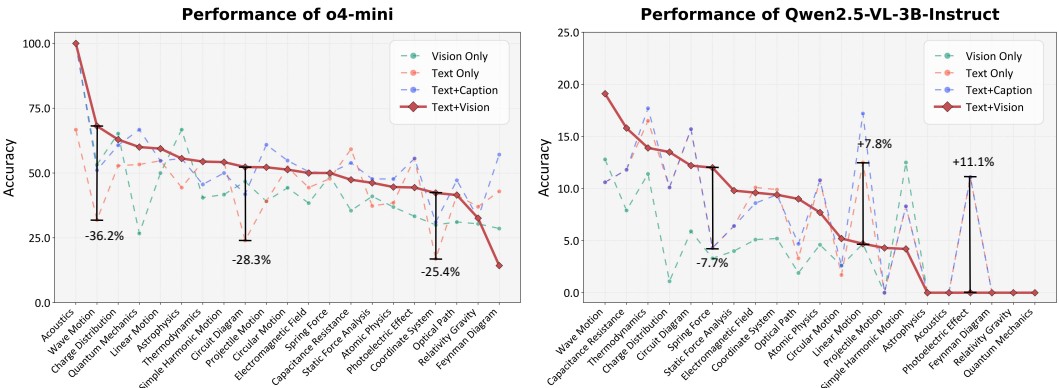

Figure 3: The sensitivity of models to different diagram types under TV/TC/TO/VO settings.

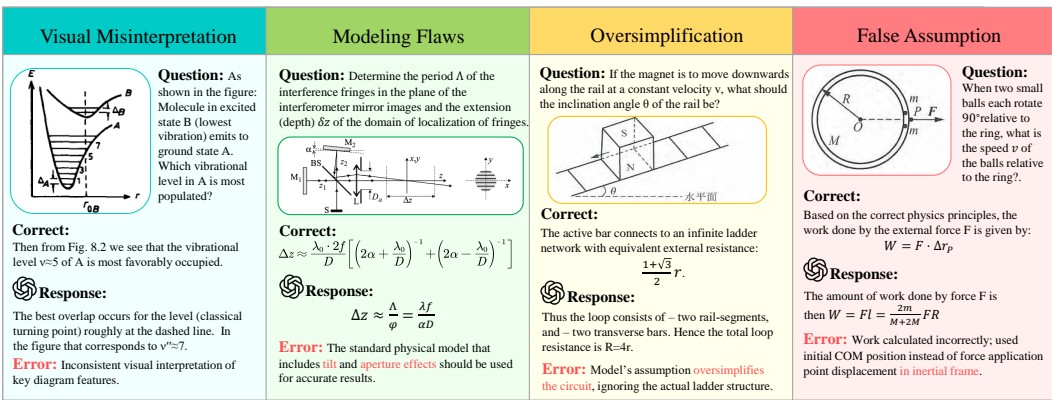

Figure 4: Examples of primary error patterns. Quantitative analyses are presented in Appendix E.

diagrams, e.g., Qwen performs better in *Circuit Diagram* than *Quantum Mechanics*, while o4-mini shows the opposite preference pattern. Unlike o4-mini, the accuracy for *Linear Motion* and *Photoelectric Effect* shows significant improvement after removing visual inputs. This suggests that models with weaker multimodal perception capabilities may misinterpret visual information, leading to poorer reasoning outcomes than random guessing based on text alone.

### 4.5 Failure Mode Analysis

Through systematic analysis of o4-mini's reasoning processes across 10% stratified samples, we identify four major error types: 1) **Visual Misinterpretation**: Persistent errors in extracting numerical values from coordinate plots, missing critical variables/symbols/units in graphical data, and flawed interpretation of geometric relationships. 2) **Modeling Flaws**: Fundamental misunderstandings in translating problem statements to physical models, including incorrect circuit schema, angular relationships in optics, and boundary conditions for dynamic systems. 3) **Oversimplification**: Neglect explicit constraints in logical deduction and omit critical computational steps. 4) **False Assumptions**: Introduction of extraneous conditions or mathematical constraints absent in original specifications, arbitrarily altering problem scope, which led to major divergence from problem statement. Notably, Visual Misinterpretation and Modeling Flaws reflects the **perceptual** and **utilization** capabilities of multimodal information, respectively. In future work, greater emphasis should be placed on enhancing the model's capacity for **fine-grained parsing** of complex images and **rule-based modeling**.

## 5 Conclusion

We introduce **SEEPHYS**, a pure multimodal benchmark for physics reasoning with 2,000 questions and 2,245 images across 8 knowledge levels, 7 core subjects with 21 distinct diagram types.

## Acknowledgements

This work is also supported by Scientific Research Innovation Capability Support Project for Young Faculty (No.ZYGXQNJSKYCXNLZCXM-I28), National Natural Science Foundation of China (NSFC) under Grants No.62476293 and General Embodied AI Center of Sun Yat-sen University. Yinya Huang is supported by an ETH AI Center Postdoctoral Fellowship.

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

# Appendix

# A Statistics

This section provides comprehensive quantitative and qualitative analyses of **SEEPHYS**'s composition. All data reflects the final curated version after expert validation and de-duplication. Table 5 presents the statistical summary of the dataset.

Our **SEEPHYS** comprises of 2,000 rigorously validated questions paired with 2,245 diagrams (averaging 1.12 images per question). The questions span 7 core physics fields and are stratified across 8 knowledge levels from middle school to PhD qualifying exams. Notably, 18.6% of problems target PhD-level reasoning, while 22.6% represent advanced Olympiad challenges. The benchmark emphasizes multimodal reasoning: 75% of questions are Vision-Essential, which necessarily requires diagram interpretation for solving (e.g., analyzing Feynman diagrams), while 25% are Vision-Optional, where visuals supplement text. Questions are language-balanced (1,039 English vs. 961 Chinese) and 88% have multi-step reasoning annotations, validated via expert annotation. Visual diversity is ensured through 21 diagram types (e.g., circuit schematics, free-body diagrams), curated by domain experts. The dataset's composition supports granular evaluation of MLLMs' physics understanding across textual, visual, and reasoning dimensions.

Figure 5: Statistics of our benchmark.

| Statistics | Number |
|---|---|
| Total Questions | 2,000 |
| Total Images | 2,245 |
| Visual Enhanced Samples | 2,000 |
| Subjects | 7 |
| Diagram Types | 21 |
| EN: CN | 1039: 961 |
| Reasoning | 88% |
| *Vision Enrichment Levels* | |
| Vision-essential | 75% |
| Vision-optional | 25% |
| *Knowledge Levels* | |
| Middle School | 5.1% |
| High School | 12.5% |
| Beginner Olympiad | 5.4% |
| Advanced Olympiad | 22.6% |
| Undergraduate | 17.8% |
| Senior Undergraduate | 11.0% |
| Master | 7.3% |
| PhD | 18.6% |

## A.1 Attributes

The following are the basic contents of the 7 covered **subjects**:

- Classical Mechanics (CM): The study of motion and forces on macroscopic objects, from linear motion, circular motion, projectile to planetary orbits.

- Electromagnetism (EM): Examines electric/magnetic fields and their interactions with matter, covering RC circuits to Maxwell's equations.

- Astrophysics, Cosmology & Gravitation (ACG): Investigates celestial phenomena, universe evolution, and gravitational interactions at all scales.

- Optics (OPT): Focuses on light behavior (reflection/refraction) and its applications in lenses, lasers, and optical technologies, this section also covers wave-related physics of acoustics.

- Atomic, Molecular, Nuclear & Particle Physics (AMONP): Studies fundamental particles and their interactions, spanning quarks to complex nuclei. It also contains emergent properties of solids/liquids and novel material design.

- Quantum Mechanics, Information & Technology (QMIT): Explores quantum systems for computing and communication applications.

- Thermodynamics & Statistical Mechanics (TSM): Analyzes energy transfer and microscopic behavior of particle ensembles.

The following are the basic contents of the 21 covered **diagram types**:

- Charge Distribution: Visualizes spatial arrangements of electric charges and their field effects.

- Feynman Diagram: Represents particle interactions through standardized symbolic notation in quantum field theory.

- Relativity and Gravity: Depicts spacetime curvature and relativistic effects near massive objects.

**Image:**

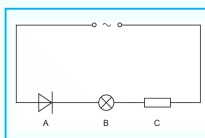

**Subject: EM**

**Level: Middle**

**Vis: Essential**

**Question:** The figure shows this power supply connected in a circuit. In each time period of the a.c., $1.5 \times 10^{17}$ electrons pass through component A. The charge on an electron is $1.6 \times 10^{-19} C$. Calculate the average current in the circuit during one time period.

**Image:**

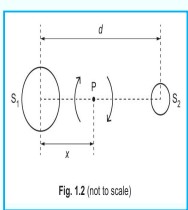

Fig. 1.2 (not to scale)

**Subject: ACG**
**Level: High**
**Vis: Essential**

**Question:** The stars $S_1$ and $S_2$ rotate with the same angular velocity $\omega$ about a point P, as illustrated in Fig. 1.2. Point P is at a distance $x$ from the centre of star $S_1$. The period of rotation of the stars is 44.2 years. By considering the forces acting on the two stars, deduce an expression for the ratio of the masses of the stars.

**Image:**

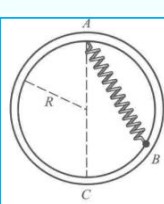

**Subject: CM**

**Level: UG**

**Vis: Essential**

**Question:** 一根原长 $l_0$的弹簧，当下端悬挂质量为 $m$ 的重物时，弹簧长 $l = 2l_0$。现将弹簧一端悬挂在竖直放置的圆环上端 $A$ 点，设环的半径 $R = l_0$，把弹簧另一端所挂重物放在光滑圆环的 $B$ 点，如图所示。已知 $AB$ 长为 $1.6R$。当重物在 $B$无初速地沿圆环滑动时，试求重物在B点的加速度。

**Image:**

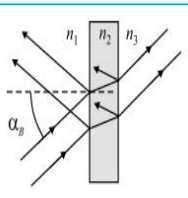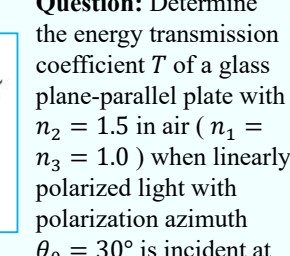

**Subject: OPT**
**Level: SUG**
**Vis: Essential**

**Question:** Determine the energy transmission coefficient $T$ of a glass plane-parallel plate with $n_2 = 1.5$ in air ( $n_1 = n_3 = 1.0$ ) when linearly polarized light with polarization azimuth $\theta_0 = 30°$ is incident at the Brewster angle $\alpha_B$.

**Image:**

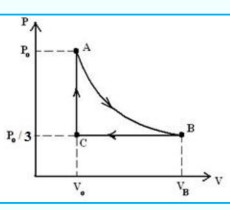

**Subject: TSM**

**Level: BO**

**Vis: Essential**

**Question:** A monatomic ideal gas undergoes the reversible cyclic process (ABCA) shown in the PV diagram. Process $A \rightarrow B$ is adiabatic. What is the efficiency of this engine?

**Image:**

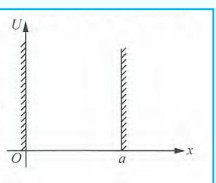

**Subject: QMIT**
**Level: AO**
**Vis: Optional**

**Question:** 势阱中的粒子不能到达 $x \leq 0, x \geq a$ 位置，粒子的势能为零，动能取为经典动能，质量记为 $m$，将氢原子中电子的能量取为经典动能与库仑势能之和，试求电子的基态（能量最低的定态）轨道半径和能量。

**Image:**

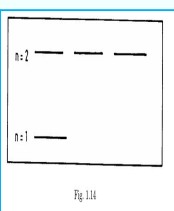

Fig. 1.14

**Subject: AMONP**

**Level: MA**

**Vis: Optional**

**Question:** Consider the ground state and $n = 2$ states of hydrogen atom. There are four corrections to the indicated level structure that must be considered to explain the various observed splitting of the levels. These corrections are: (a) Lamb shift, (b) fine structure, (c) hyperfine structure, (d) relativistic effects. Which of the above apply to the $n = 2$, $l = 1$ state? Answer in the name of the corrections.

**Image:**

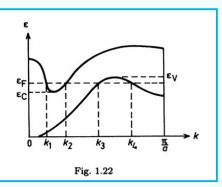

Fig. 1.22

**Subject: AMONP**
**Level: PhD**
**Vis: Essential**

**Question:** Figure shows an energy versus wave vector diagram for electrons in a one-dimensional solid. If $n$ is the number density for electrons and $p$ is that for holes, what can be inferred about $p/n$?

Figure 6: Cases of our **SEEPHYS**.

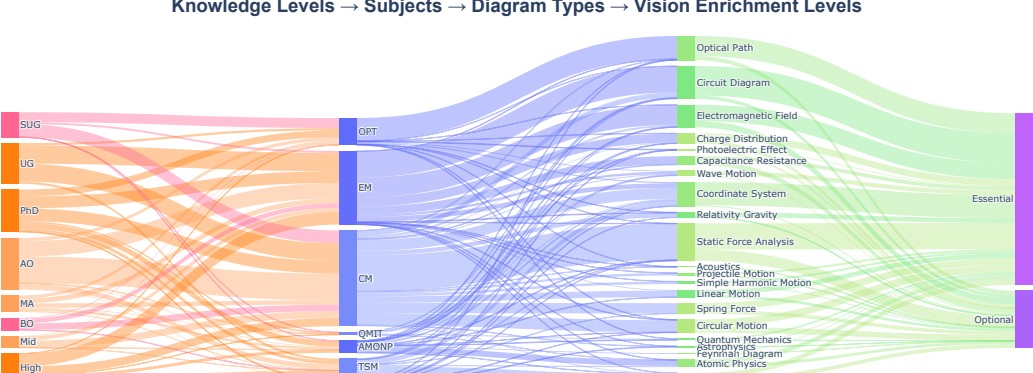

Figure 7: The Distribution of knowledge levels, subjects, diagram types, and vision enrichment levels.

- Atomic Physics: Illustrates atomic energy levels, transitions, and spectral phenomena.
- Static Force Analysis: Demonstrates equilibrium conditions through free-body diagrams and force vectors.
- Photoelectric Effect: Shows electron emission processes under photon irradiation with energy thresholds.
- Linear Motion: Characterizes one-dimensional kinematics with position-time/velocity-time graphs.
- Coordinate System: Provides reference frames for analyzing physical quantities in 2D/3D space.
- Astrophysics: Models celestial phenomena like stellar evolution or orbital mechanics.
- Spring Force: Displays Hooke's law applications and oscillatory systems with restoring forces.
- Optical Path: Traces light propagation through media with reflection/refraction principles.
- Simple Harmonic Motion: Visualizes periodic motion through phase-space plots or pendulum dynamics.
- Quantum Mechanics: Represents wavefunctions, potential wells, and quantum superposition states.
- Circular Motion: Analyzes centripetal forces and angular kinematics in rotational systems.
- Thermodynamics: Charts thermodynamic cycles, heat engines, and entropy changes.
- Acoustics: Demonstrates sound wave propagation, interference, and standing wave patterns.
- Circuit Diagram: Standardized schematics for electrical networks with component symbols.
- Projectile Motion: Parabolic trajectories under uniform gravity with drag effects.
- Wave Diagram: Graphical representations of wavelength, frequency, and wave interference.
- Electromagnetic Field: Maps field lines and flux distributions in electric/magnetic systems.
- Capacitance Resistance: Characterizes RC circuits with charge/discharge time constants.

Figure 6 shows samples of our SEEPHYS. Moreover, we present the distributions of knowledge levels, subjects, diagram types, and vision enrichment levels in Figure 7.

## A.2 Data Source

We comprehensively collect visual Physics problems from existing public question repositories:

**Master's and PhD.** We select questions with Master's and PhD qualifying exams level from Major American Universities Ph.D. Qualifying Questions and Solutions. This collection comprises problems from graduate-school entrance and qualifying examinations at seven major U.S. universities,

spanning seven volumes: Mechanics, Electromagnetism, Optics, Atomic, Nuclear and Particle Physics, Thermodynamics and Statistical Physics, Quantum Mechanics, and Solid State Physics. The series is distinguished by its comprehensive coverage, with problems that span a wide spectrum of topics within each area and frequently overlap multiple areas. These problems are notable for their versatility in applying physical laws and principles to up-to-date, realistic situations, while often requiring minimal complex mathematical manipulation. They effectively blend the objectives of enhancing the understanding of physical principles with the ability for practical application.

**Undergraduate.** This collection includes College Physics, General Physics, Theoretical Mechanics and Wave Optics. College Physics designs for students in science and engineering disciplines at general higher education institutions, offering broad coverage and a combination of varying difficulty levels. General Physics covering a wide spectrum of foundational university physics, this collection focuses on the analysis of problem-solving strategies and the application of fundamental methods, often presenting multiple solution approaches. Theoretical Mechanics covers all the teaching contents of theoretical mechanics and includes highly specific exercises, emphasizing the techniques for solving practical problems using general theorems and methods. Wave Optics presents problems related to a wide scope of wave phenomena in optics, studied within the framework of the university course of general physics. Largely associated with visual-spatial perception.

**Olympiad competitions.** The International Physics Olympiad (IPhO) is a premier global competition featuring problems of exceptional difficulty and innovative conceptual design, spanning mechanics, electromagnetism, thermodynamics, optics, and modern physics. Its challenges emphasize multidimensional problem-solving, requiring participants to synthesize physical principles in non-traditional contexts, such as astrophysical systems or nanotechnology. Problems often test advanced mathematical techniques, including tensor analysis in continuum mechanics, and demand critical modeling skills, such as dimensional analysis or symmetry-based simplifications. The Chinese Physics Olympiad (CPhO), renowned for its theoretical rigor and computational intensity, integrates calculus deeply into physics problem-solving, employing methods like variational principles in constrained dynamics. Its multi-stage problems frequently involve layered complexities, for instance, incorporating relativistic corrections in electromagnetic boundary-value problems, under strict time constraints. The CPhO's quality aligns with the IPhO, with some problems exceeding its difficulty, making it one of Asia's most demanding competitions.

**Middle and high school.** Past examination papers from the Cambridge Assessment International Education for IGCSE Physics and AS & A-Level Physics constitute this source. The data quality is high, reflecting a well-established and internationally recognized curriculum. These problems are characterized by their structured approach to assessing physics knowledge and understanding, ranging from fundamental concepts at the IGCSE level to more advanced topics in the AS & A-Level, with some questions incorporating elements of undergraduate-level content. The questions emphasize conceptual clarity, data interpretation, and the application of physics principles to varied contexts.

### A.3 Annotators Information

For data annotation and evaluation, we recruit 7 annotators from engineering and physics programs, consisting of 4 undergraduates, 3 PhD candidates. All annotators demonstrated strong competencies in both secondary and tertiary-level physics through rigorous qualification assessments. One undergraduate student and one PhD candidate, both highly familiar with all knowledge levels covered by this benchmark, conduct a professional secondary review of the annotation results. Since all annotators are coauthors of this study, they are sufficiently motivated to participate in the annotation task and do not require additional compensation or benefits. Furthermore, this research is dedicated to academic AI evaluation and does not involve recruiting human subjects, making it exempt from Institutional Review Board (IRB) regulations. As part of the institutional research activities, all data used in this work were obtained from publicly available and legally permissible sources, with no collection of private or protected sensitive demographic information.

# B  Limitations

## B.1  Process Reward

Many current models are capable of generating responses that include intermediate explanatory steps, which may reflect their internal logical reasoning patterns. This is a valuable, refined evaluation of LLM physics reasoning. However, due to the high cost of process annotation and the inherent uncertainty in evaluation (intermediate results can be expressed in multiple ways, and some problems may have multiple valid solutions), this study so far provides outcome-based reward signals. Future work should focus on improving the reliability of process evaluation and integrating it with outcome accuracy to design a comprehensive metric for assessing reasoning capabilities.

## B.2  Low-Resource Evaluation Method

Although SymPy is partially employed for quick result matching, the evaluation pipeline in this work still primarily relies on LLMs to provide reward signals. It is because **SEEPHYS** encompasses diverse open-ended question types (e.g., computation, derivation, case-based analysis) with inherent uncertainty in model output formats. Only a small fraction of responses could be directly verified using automated tools, resulting in a resource-intensive evaluation process that hinders broader adoption in the research community. Future work should focus on designing more efficient and accurate rules or tools for assessing open-ended question answers.

## B.3  Connection between Theory and Real-World Scenarios

The questions used in this benchmark are sourced exclusively from existing theoretical physics databases, covering primarily high-level concepts and principles in the physics discipline, with minimal inclusion of engineering-related problems (e.g., architecture, mechanical engineering, and biomechanics) or cross-modal sensory problems that better approximate real-world applications. Future research should further examine the relationship between a model's theoretical reasoning and its ability to model real-world phenomena—referred to as world modeling capability.

# C  Data Collection Pipeline

## C.1  Collection

Our **SEEPHYS** benchmark aggregates educational materials (textbooks, exercises, exams, and contest problems) from globally distributed education systems, covering East Asian, European, North American, and other regional curricula. To preserve authentic multilingual evaluation, we retain all source languages without translation, maintaining a 961:1039 Chinese-English text ratio. The corpus comprises 7,000+ PDF pages processed through Mathpix's OCR system to generate structured Markdown representations.

Each acquired question must satisfy the following criteria:

- Vision Information Enrichment: For Vision-Essential subset, selected images should contain essential information for problem-solving. Diagrams or illustrations should be non-decorative and directly support the question's resolution. For Vision-Optional subset, images should not contain essential problem-solving information (e.g., numerical values) and should serve only as supplementary visual cues.

- Knowledge Spectrum: The content should cover topics ranging from middle school to PhD qualifying exam levels.

- Without Ambiguity: Only questions with definitive answers are included, while open-ended questions permitting multiple interpretations are excluded. Questions requiring explanatory answers longer than three sentences are discarded.

Since the collected questions may contain grammatical or formatting errors after OCR processing, we employ the prompt to guide GPT-4.1 in performing preliminary linguistic correction (Figure 9).

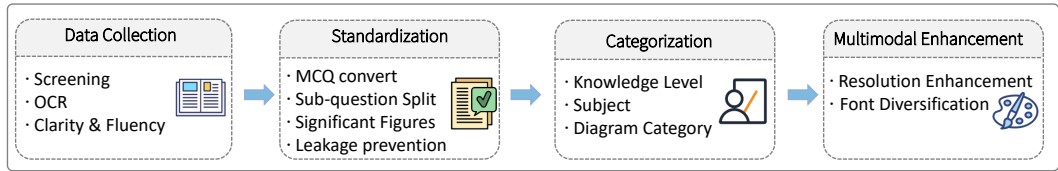

Figure 8: Overview of the data collection pipeline.

## C.2 Standardization

Many source materials (particularly textbook exercises and competition problems) contain compound questions comprising multiple independent sub-questions (e.g., "Prove X and then calculate Y"). So we systematic decomposition of compound questions into atomic units and then reconstruct them with shared contextual elements when logically dependent. It ensures each question in our dataset represents a single, self-contained cognitive task while preserving original problem relationships through metadata tagging. To modify multi-choice questions to open-ended questions, we develop stem rewriting to remove choice-specific references (e.g., changing "Which of the following" to "Determine"). For computational problems, we address a significant figures annotation based on problem constraints. This approach reduces false negatives in automated scoring while accommodating legitimate solution variants.

To prevent data leakage, we implemented a dual-phase verification protocol: 1) Systematically use and disabling GPT-4o's web search functionality via API parameters to eliminate questions exhibiting accuracy fluctuations. 2) Manual Google verification of all correctly answered items.

Our two-phase validation protocol ensures conceptual integrity:

- Primary annotation by domain experts.

- Cross-validation by secondary annotators.

- When the two annotators disagree in their judgment regarding physics concepts, a third arbitrator holding a PhD in physics is engaged to conduct the final review.

- Continuous validation sampling (10% of processed questions) throughout dataset development

## C.3 Categorization

As all source materials originate from discipline-specific examinations, we initially classify questions into 7 broad thematic categories based on their subject matter. To analyze LLMs' sensitivity to visual elements, we implement a fine-grained classification system comprising 21 distinct diagram types. Notably, coordinate systems are treated as composite categories, as they may incorporate multiple graphical components across different subject domains. Through comparative analysis of international curricula standards and expert deliberation, we establish an 8-tier knowledge hierarchy. Olympiad competition problems are split into beginner and advanced tiers based on average accuracy rates, while undergraduate-level questions are divided into undergraduate (non-mathematical physics) and senior undergraduate (mathematical physics) categories.

## C.4 Multimodal Enhancement

We also provide a pure multimodal subset containing 2,000 composite image examples. Each example consists of a single image integrating both textual and graphical elements. We first generate detailed captions for each sample using o4-mini, which include comprehensive descriptions of geometric features and numerical data through the prompt template shown in Figure 10. Subsequently, we render each question with its corresponding diagram into a composite image under 4096×4096 pixels resolution. The rendering process incorporates varied font types and sizes, while dynamically adjusting text-to-diagram spacing based on each chart's maximum dimensions to ensure optimal layout compactness. Cases are shown in Figure 11.

You are a physicist, please enhance physics text with the following instructions:

**1. STRUCTURE:**
  - Merge broken paragraphs
  - Remove redundant line breaks
  - Fix string omissions
  - Preserve LaTeX math: $E=mc^2$
  - Correct LaTeX syntax errors

**2. TERMINOLOGY:**
  - Standardize terms ("Kirchoff"→"Kirchhoff")
  - Keep glossary: {"电容":"capacitance"}

**3. FLUENCY:**
  - Fix grammar/syntax errors
  - Clarify ambiguous phrases

**4. VALIDATION:**
  - Verify numbers/units unchanged
  - Flag uncertain conversions

**Example:**
Input: "eltric field E=kq/r²\\n\\nwhere k is\\nCoulomb const"
Output: "Electric field $E=\frac{kq}{r^2}$ where $k$ is Coulomb's constant"

Figure 9: Instruction for OCR Text Enhancement with GPT-4.1.

## D   Experimental Settings

### D.1   Models

In our experiments, we evaluate the performance of several state-of-the-art and representative LLM-s/MLLMs. For LLMs, we provide text-based prompts in the form of "question + caption" to guide the models in generating answers. For MLLMs, general-purpose models capable of processing interleaved image-text sequences are tested on the full benchmark. For most open-source models, we use the hyperparameter torch.dtype=torch.float16. We set temperature=0, with a maximum token limit of 8192 and a maximum image resolution of 4096×4096 pixels. Additionally, for other parameter configurations, we generally follow the settings provided in the original papers, their code repositories, or Hugging Face's example configurations. The language models used in this study are briefly described as follows:

- **DeepSeek-R1:** It is based on a four-stage training process incorporating Supervised Fine-Tuning and Reinforcement Learning. Despite utilizing only minimal annotated data, it significantly enhances the model's reasoning capabilities. In tasks such as mathematics, coding, and natural language reasoning, the model, with 670 billion parameters, achieves performance comparable to OpenAI's o1 official version.

- **DeepSeek-V3:** It is a powerful Mixture of Experts (MoE) language model, activating approximately 37 billion parameters per token. DeepSeek-V3 pioneers an auxiliary-loss-free load balancing strategy and incorporates a multi-token prediction training objective to achieve enhanced performance.

Figure 10: Instruction for Diagram Caption with o4-mini.

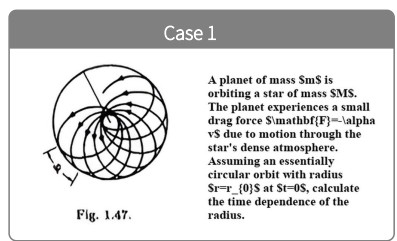 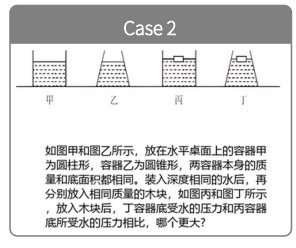 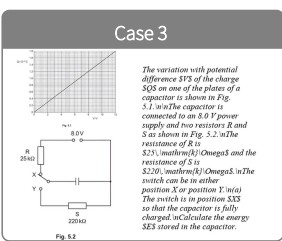

Figure 11: Cases of pure multimodal subset.

- **Qwen3-235B-A22B:** The model has 235 billion parameters, activates 22 billion parameters per inference, and consists of 128 experts, with 8 activated during each forward pass. This design significantly enhances computational efficiency and scalability while maintaining high performance.

- **Qwen2.5-72B-Instruct:** This is a dense, decoder-only language model pre-trained on a dataset of up to 18 trillion tokens. Qwen2.5-72B-Instruct supports context lengths of up to 128K tokens and can generate content with a maximum length of 8K tokens.

- **QwQ-32B:** QwQ-32B employs reinforcement learning techniques, supporting the visualization of the model's reasoning process and a context length of 131,072 tokens. It is capable of solving advanced mathematical problems, including algebra, geometry, calculus, and more.

- **R1-Distilled-Llama-70B:** Built upon the Llama-3.3-70B-Instruct model, it has been meticulously fine-tuned using DeepSeek R1's outputs, enabling outstanding performance across multiple benchmarks. While maintaining low costs, its capabilities rival those of larger, cutting-edge models.

- **Llama-4-Scout-17B:** This is the latest general-purpose multimodal model in the Llama series, featuring 16 expert modules, 17 billion active parameters, and a total of 109 billion parameters.
- **Gemma3-27B:** It is developed by the Google DeepMind team and incorporates several enhancements based on Gemma 2, including the addition of visual comprehension capabilities, support for more languages, and the ability to process contexts of up to 128K tokens.
- **Llama-3.1-8B:** The Llama 3.1 model features a 128K context length and is optimized for scenarios with limited computational resources.

The multimodal language models used in this study are briefly described as follows:

- **OpenAI o4-mini:** OpenAI o4-mini is a compact model optimized for fast, cost-efficient inference. Despite its reduced size and lower cost, it delivers exceptional performance in math, coding, and vision tasks, while maintaining high throughput.
- **OpenAI o3-mini:** The o3-mini demonstrates exceptional performance in STEM reasoning tasks. It achieves comparable results to the o1 model in mathematics, programming, and scientific tasks with significantly faster response times.
- **OpenAI o1:** o1 is a cutting-edge model released by OpenAI specifically designed for complex reasoning tasks, trained using reinforcement learning. The model is capable of engaging in prolonged deliberation before providing answers, and its performance empirically validates the existence of test-time scaling laws.
- **Gemini-2.5-Pro:** It is a hybrid reasoning model proposed by Google DeepMind, supporting native multimodal capabilities and a 1 million token context window, achieving significant advancements in coding, reasoning, and multimodal tasks. In this paper, we use Gemini-2.5-Pro-Exp-03-25.
- **Claude 3.7 Sonnet:** It is Anthropic's most advanced large language model and the first to combine multiple reasoning approaches. Claude 3.7 Sonnet can both provide quick answers and engage in deeper, step-by-step thinking—with the entire reasoning process visible to users.
- **Doubao-1.5-pro:** It adapts a sparse MoE architecture, maintaining a training-inference co-design approach from the pre-training phase. With only a small fraction of activated parameters, it outperforms massive dense pre-trained models like Llama3.1-405B.
- **GPT-4.1:** The model comprehensively surpasses GPT-4o and GPT-4o mini in coding, instruction following, and long-context understanding, while being more cost-effective, faster, and capable of processing contexts up to 1 million tokens.
- **GPT-4o:** This model is trained on text, visual, and audio data. Its unified approach ensures that all inputs—whether text, images, or sound—can be processed simultaneously by a single neural network.
- **QvQ-72B-preview:** It is an open-source multimodal reasoning model developed by Qwen team, with a special focus on enhancing visual reasoning capabilities. It supports extracting precise information (e.g., object height, quantity) from images and can interpret the deeper meaning behind pictures.
- **Qwen-VL series:** The Qwen2-VL series models employ a three-stage fine-tuning approach to sequentially train different modules. It utilizes naive dynamic resolution mechanism and multimodal rotary position embedding to effectively fuse information from text, images, and videos of varying scales. Qwen2.5-VL series implements window attention, which boosts both training and inference speeds while significantly enhancing general image recognition capabilities.
- **Llama-3.2-Vision series:** Llama 3.2-Vision series is a collection of pre-trained and finetuning vision-language models that support text + image inputs with text-only outputs, featuring a 128K context length.
- **LLaVA-NeXT-7B:** This model is designed to improve image-text interaction capabilities, particularly in OCR (Optical Character Recognition) and commonsense reasoning. It employs Vicuna-7B as its language model and significantly boosts visual reasoning performance through dynamic high-resolution input processing and an optimized visual instruction-tuning dataset.

- **Phi-4-multimodal:** Phi-4-Multimodal is a 5.6B-parameter multimodal model that integrates text, visual, and speech/audio input modalities. It employs an modality expansion approach, utilizing LoRA adapters and modality-specific routers to enable interference-free combination of diverse modalities during inference.

- **InternVL2.5-8B:** InternVL2.5-8B integrates the pre-trained InternViT-300M vision backbone with large language models (InternLM 2.5) through a randomly initialized 2-layer MLP projector. The model additionally introduces native support for high-resolution multi-image inputs.

- **LLaVA-OneVision-7B:** It adopts Qwen-2 as its LLM backbone and SigLIP as the visual encoder, with the two modules connected via a parameterized 2-layer MLP. This architecture achieves state-of-the-art performance for open-source multimodal large models across single-image, multi-image, and video tasks.

## D.2 Environment

We deploy advanced reasoning models with a computational infrastructure. we use a Linux-based environment equipped with CUDA-enabled GPUs (8 * 80G NVIDIA A800) to accelerate tensor operations. The software stack includes PyTorch 2.5.1 with CUDA 12.4 support, alongside Python 3.10 for compatibility with modern machine learning libraries. For all the models, half-precision (FP16) quantization is enabled to optimize runtime.

## D.3 Inference Template

During the experiments, we design efficient Chain-of-Thought (CoT) templates to enhance the model's reasoning capabilities. Given that physics problems often involve approximate calculations, we incorporate significant figure hints in the input. As shown in the Figure 12, we provide customized prompts in both English and Chinese to accommodate different linguistic contexts.

---

**Inference Templates**

**English:**
<image>
Please answer this question with reasoning. First output your reasoning process in
<think> </think> tags and then output the final answer in <answer> </answer> tags.

The final answer should retain {x} significant figures.

**Chinese:**
<image>
请用推理来回答这个问题。首先在<think></think>标签中输出推理过程，然后在
<answer></answer>标签中输入最终答案。

最终答案应保留{x}位有效数字。

---

Figure 12: English/Chinese template for inference.

## D.4 Evaluation

During the evaluation phase, we integrate automated verification with LLM-as-judge method to generate comprehensive reward signals. The assessment pipeline first leverages SymPy for rapid mathematical matching between model responses and ground truth. Samples failing this validation are then subjected to secondary scoring by LLM, ensuring robust evaluation coverage across all response types. Given the inherent complexity of physics reasoning tasks, the LLM's judging process

is implemented as a two-stage pipeline consisting of answer extraction followed by scoring. The first stage involves guiding the model to extract clean answers by removing extraneous characters, identifying numerical values and units, and handling cases with multiple valid answers. The second stage requires model to perform precise unit conversions and recognize various mathematically equivalent expressions when applying the scoring criteria. We calibrate these pipeline using carefully designed few-shot prompts as illustrated in Figure 13 and Figure 14. Through manual verification of 200 samples, the DeepSeek-V3 model demonstrates reliable judging capability with an error rate below 5%, validating the robustness of this evaluation methodology for complex physics reasoning tasks.

---

**Answer Extracting Prompt**

Please read the following example. Then extract the answer from the model response and type it at the end of the prompt.

**Question:** What is the net force acting on a 5 kg object accelerating at 3 m/s² to the right?
**Model Answer:** Using F = ma, the net force is 15 N to the right.
**Extracted Answer:** the net force is 15 N to the right.
...

**Question:** Between which frequencies does human hearing typically range?
**Model Answer:** Human hearing ranges between 20 Hz and 20,000 Hz.
**Extracted Answer:** [20 Hz, 20000 Hz]

Now please extract the answer, DONNOT output explanation:
**Question:** {question}
**Model Answer:** {model answer}
**Extracted Answer:**

Figure 13: Prompt for answer extracting.

# E   More Experiments

## E.1   Comparison of Theory Memorization and Problem-Solving Skills

Since the difficulty level labeling can be easily influenced by the subjective judgment of human annotators, SeePhys does not directly provide related tags (i.e., problem-solving skills level). We categorize the questions into 8 levels based on the knowledge content involved (i.e., conceptual level of theory needed as classified by incremental grades). Their order corresponds to the knowledge content (by grade distribution) rather than the actual difficulty. We placed Olympiad competitions between high school and undergrad since competitions like the IMO usually touch broader coverage than high school but does not require higher mathematics skills such as Calculus.

However, in order to further compare the **Theory Memorization** and **Problem-solving Skills** of SOTA models, we first reclassified the knowledge levels and the aggregated evaluation results are shown in Table 4. To minimize annotators' subjective bias regarding difficulty levels as much as possible, we also applied majority voting in Table 5 to recalculate the accuracy of problems with different difficulty levels. It is found that models like Gemini-2.5-Pro excel in problem-solving tasks but show weaker high-level theory retention (44.2% in PhD, less than o4-mini). It proves knowledge depth does not guarantee strong problem-solving. In Table 2, we observe a significant imbalance in Doubao-1.5-pro's theoretical memorization capabilities. While it demonstrates outstanding performance in memorizing middle school level knowledge (70.6%), it exhibits clear deficiencies in mastering higher-level concepts.

Figure 14: Prompt for answer scoring.

Table 4: Comparison of Theory Memorization and Problem-solving Skills across different difficulty levels.

| Models | Theory Memorization | | | | Problem-solving Skill | | | |
|---|---|---|---|---|---|---|---|---|
| | UG | SUG | MA | PhD | Mid | High | BO | AO |
| Gemini-2.5-Pro | **64.2** | **50.2** | **53.8** | 44.2 | 69.6 | **66.7** | **64.5** | **46.7** |
| o4-mini | 53.8 | 45.7 | 51.0 | **53.4** | 66.7 | 61.8 | 56.1 | 41.8 |
| Doubao-1.5-pro | 56.6 | 34.7 | 40.7 | 37.5 | **70.6** | 58.2 | 49.5 | 29.2 |

### E.2 Statistical Analysis of Failure Reasonings

To provide the community with quantitative error analysis results, we conduct manual inspection of 100 error samples common to o4-mini, Gemini-2.5-Pro, and Qwen2.5-VL-3B. We shows 9 different error patterns in Table 6. First, all three models exhibit significant Modeling Flaws (e.g., incorrect theorem applications and formula misuse), while demonstrating relatively fewer Text Misinterpretation and Numerical Miscalculation errors. This suggests that even weaker models have acquired fundamental text recognition and numerical computation capabilities, yet still show substantial gap in applying principles of physics. Second, Visual Misinterpretation emerged as the second most frequent error pattern, indicating persistent weaknesses in multimodal comprehension. Error frequencies for Overthinking and Oversimplification show notable variation across models.

Table 5: Extended comparison of Theory Memorization and Problem-solving Skills across different knowledge and difficulty levels. We use majority voting approach across five models (Gemini-2.5-Pro, o4-mini, Doubao-1.5-pro, Qwen2.5-VL-72B-Inst and QVQ-72b-preview) to determine difficulty labeling, resulting in six difficulty level tags.

| Theory Memorization | Mid | High+BO+AO | UG | SUG | MA | PhD |
|---|---|---|---|---|---|---|
| Gemini-2.5-Pro | 69.6 | 52.1 | 64.2 | 50.2 | 53.8 | 44.2 |
| o4-mini | 66.7 | 48.4 | 53.8 | 45.7 | 51.0 | 53.4 |
| Doubao-1.5-pro | 70.6 | 42.8 | 56.6 | 34.7 | 40.7 | 37.5 |
| **Problem-solving Skill** | **100% (125)** | **80% (247)** | **60% (342)** | **40% (346)** | **20% (362)** | **0% (578)** |
| Gemini-2.5-Pro | 100 | 96.0 | 90.4 | 67.1 | 39.5 | 0 |
| o4-mini | 100 | 96.8 | 92.1 | 65.6 | 34.5 | 0 |
| Doubao-1.5-pro | 100 | 97.2 | 90.1 | 49.4 | 16.0 | 0 |

Table 6: Error patterns comparison of o4-mini, Gemini-2.5-Pro and Qwen2.5-VL-3B. We identify the following error patterns in the models' outputs: VM: Visual Misinterpretation; TM: Text Misinterpretation; MF: Modeling Flaws; FA: False Assumption; NM: Numerical Miscalculations; OS: Oversimplification; SM: Summarization Mistakes; OT: Overthinking; RO: Repetitive Output

| Models | VM | TM | MF | FA | NM | OS | SM | OT | RO |
|---|---|---|---|---|---|---|---|---|---|
| o4-mini [30] | 15 | 1 | 61 | 8 | 3 | 6 | 3 | 3 | 0 |
| Gemini-2.5-Pro [9] | 17 | 2 | 49 | 13 | 3 | 0 | 4 | 12 | 0 |
| Qwen2.5-VL-3B [3] | 11 | 0 | 48 | 8 | 0 | 4 | 3 | 5 | 21 |

Particularly noteworthy is Qwen2.5-VL-3B's high rate of Repetitive Output (21%), which is absent in the cutting-edge proprietary models. We attribute this to the model's limited 3B parameter scale, which likely constrains its instruction-following capacity.

### E.3 Case Studies of Failure Patterns

We also present concrete case studies illustrating common error patterns observed in outputs of o4-mini and Gemini-2.5-Pro:

# Visual Misinterpretation

**Question:** The potential varies along the *Ox* axis as shown in the figure. Determine the *x*-component of the electric field intensity for the potential distribution in the interval *a*-*b* (neglecting the situation at the endpoints of the interval).

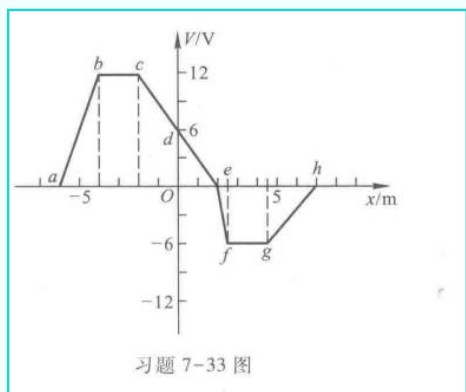

习题 7–33 图

**Response:**

...potential from 0 linearly rises to 12 V, in the x direction correspondingly changes from $x \approx -10$ m to $x \approx -5$ m, distance $\Delta x = 5$ m, therefore in this interval $dV/dx \approx (12 - 0)/5 = 2.4\ V/m$...

**Error:** It misreads the x-axis scale, incorrectly taking $\Delta x = 5m$ instead of the actual $\Delta x = 2m$ (-6 m to -4 m). This leads to an underestimated potential gradient and thus a wrong $E_x$ value.

Figure 15: Visual Misinterpretation on o4-mini.

# Visual Misinterpretation

**Question:** For the circuit shown, the three light bulbs have identical resistance *R*, the battery is ideal, and all wires have no resistance. Which light bulbs become dimmer or go out completely when the switch *S* in the circuit is closed?

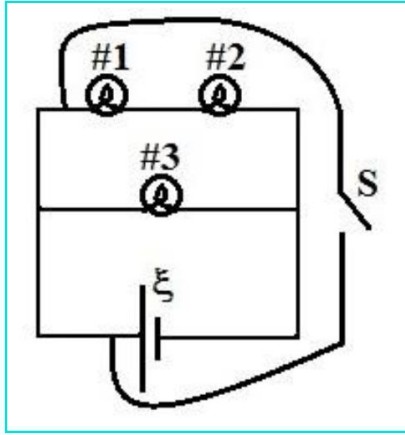

**Response:**

...once S is closed... The mid-rail bulb #3 is now shunted on its right side by a zero-ohm path (the right-rail). In effect, the voltage difference across #3 collapses to zero and all of the current chooses the short rather than flowing through #3. → #3 goes out completely.

**Error:** It **misinterprets the circuit diagram** and thinks that bulb #3 is shorted when S closes. In fact the new wire doesn't connect to the negative terminal, so bulb #3 remains lit with full voltage.

Figure 16: Visual Misinterpretation on o4-mini.

## Visual Misinterpretation

**Question:** Polonium-211 decays by alpha emission to form a stable isotope of lead **Pb**. The variation with time $t$ of the number of unstable nuclei $N$ in a sample of polonium-211 is shown in Fig. 9.1. At time $t = 0$, the sample contains only polonium-211. Use the figure to determine the decay constant $\lambda$ of polonium-211. Give a unit with your answer.

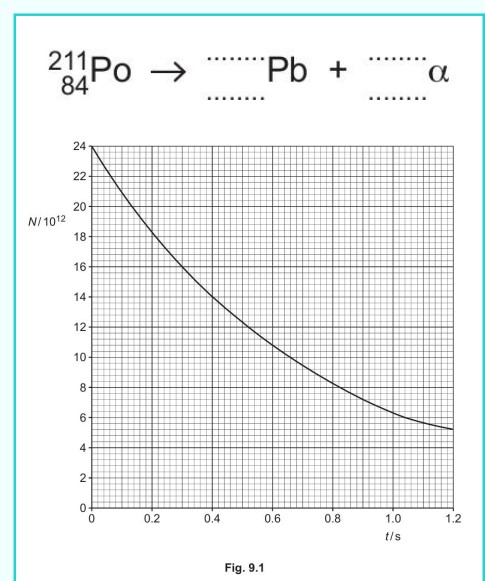

$$^{211}_{84}\text{Po} \rightarrow \;^{.......}_{.......}\text{Pb} + \;^{.......}_{.......}\alpha$$

Fig. 9.1

### Response:

At $N = 12 \times 10^{12}$ the time $t$ is approximately 0.51s. So, $T_{0.5} \approx 0.51$s.

**Error:** The actual half-life shown in the diagram is 0.52 s, while Gemini identified it as 0.51 s.

Figure 17: Visual Misinterpretation on Gemini-2.5-Pro.

# Visual Misinterpretation

**Question:** On a frictionless horizontal surface, two bodies make a head-on collision and stick together. Body 1 has a mass of 3.50 kg and initially moves to the right with speed 7.0 $m/s$. Body 2 initially is at rest. What is the mass of Body 2?

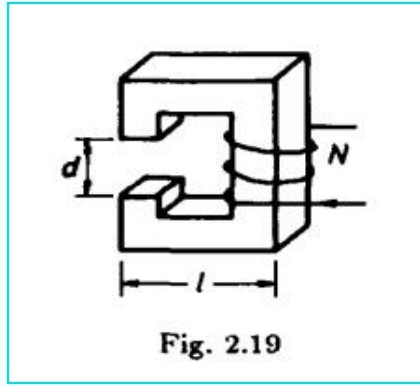

Fig. 2.19

**Response:**

Now we need to determine the mean path length in the iron, $l_{iron}$. The problem states that "the length of each side of the 'C' is $l$". A C-shape can be considered to have three main segments: a back segment and two arm segments. If the mean length of each of these three segments is $l$, then the total mean path length in the iron is: $l_{iron} = 3l$.

**Error:** It incorrectly models the total core path length as $3l$ when it should actually be $4l - d$.

Figure 18: Visual Misinterpretation on Gemini-2.5-Pro.

# Visual Misinterpretation

**Question:** A student designs the circuit shown in the figure in order to indicate changes in temperature of the thermistor T. At a temperature of 13℃, the resistance of the thermistor T is 230$k\Omega$. Calculate the potential difference measured with the voltmeter.

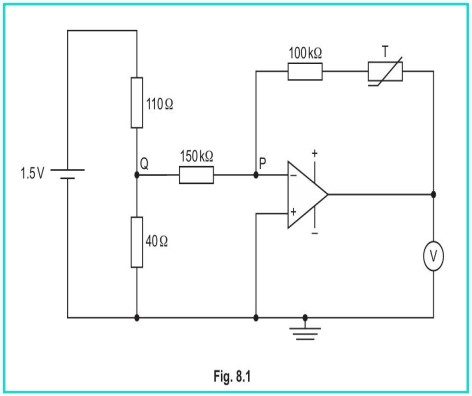

Fig. 8.1

### Response:

For an inverting amplifier, the output voltage is given by $V_{out} = -\frac{R_f}{R_{in}}V_{in}$, where $V_{in} = V_Q$ and $R_{in} = 150k\Omega$. $V_{out} = -2.2 \times 0.40V = -0.88V$.

**Error:** Misidentifying a non-inverting amplifier as an inverting amplifier when determining the circuit type, resulting in incorrect formula signs.

Figure 19: Visual Misinterpretation on Gemini-2.5-Pro.

# Modeling Flaws

**Question:** If your eyes are near-sighted and can focus an object 20 cm away without using any glasses, estimate the required diameter of the hole through which you would have good sight for objects far away.

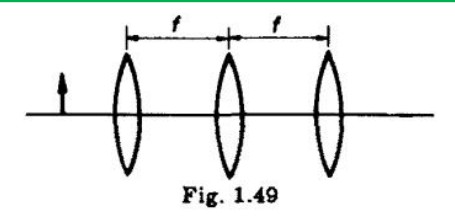

Fig. 1.49

## Response:

…..From the person's far point of 20 cm we can deduce that the unaccommodated focal length f∞ must satisfy ... $s_o$=0.20 m and $s_i \approx$ the lens–retina spacing $s_r$. A typical $s_r \approx$17 mm. hence $1/f_\infty$=1/0.20–1/0.017⇒$f_\infty$≈18.6 mm.

**Error:** It incorrectly uses subtraction when calculating the eye's unaccommodated focal length ($f_\infty$). For the near-sighted eye's far point (object at 20 cm, image on the retina at ≈17 mm), the lens formula $1/f_\infty$=1/$s_o$+ 1/$s_i$ requires addition (1/0.20+1/0.017), not the subtraction (1/0.20–1/0.017) used in the text.

Figure 20: Modeling Flaws on o4-mini.

## Modeling Flaws

**Question:** As shown in the figure, a transparent material sphere with radius $R$ has diameter $AB$. Only the regions near points $A$ and $B$ on the sphere's surface are transparent. Inside the sphere at point $B$, there is a small luminous bulb. If the refractive index of the sphere is $n$, and the image is formed on the right side of the sphere, what is the lateral magnification of the image?

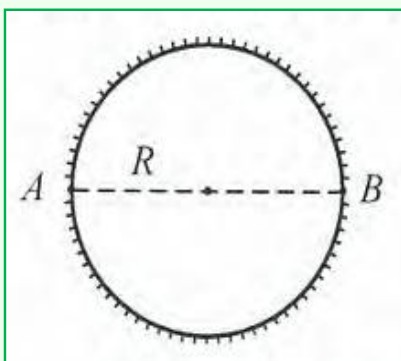

### Response:

Using the spherical imaging formula $\frac{n_1}{u} + \frac{n_2}{v} = \frac{n_2 - n_1}{R}$. Substituting gives $\frac{n}{2R} + \frac{1}{v} = \frac{1-n}{R}$ (Object distance $u = 2R$ is also stated and used as a positive value.

**Error:** It incorrectly treats the object distance $u = 2R$ as positive and the radius of curvature $R_{surf}$ as positive. It leads to an expression for image position v that differs from standard derivations using conventions like the Cartesian one.

Figure 21: Modeling Flaws on o4-mini.

# Modeling Flaws

**Question:** Five identical light bulbs are connected into a circuit as shown. All wires are ideal with no resistance, and the ideal battery has emf $\xi$. When the switch S in the circuit is closed, aside from bulb #5, which bulbs brighten?

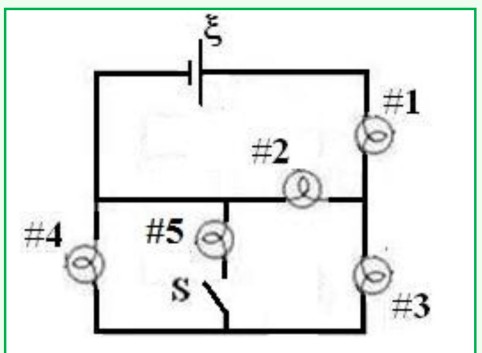

**Response:**

Bulbs #2 and #3 are in series, $R_{23} = 2R$.The combination of ( #2 and #3 ) is in parallel with the combination of ( #4 and #5 ). The equivalent resistance of this parallel part is $R'_{parallel} = \frac{R_{23} \cdot R_{45}}{R_{23}+R\_45} = \frac{2R}{5}$....Current through the series combination of #2 and #3 is $I_{23_{closed}} = \frac{V'_{parallel}}{R_{23}} = \frac{\xi}{7R}$. So, $I_{2_{closed}} = I_{3_{closed}} = I_{23_{closed}} = \frac{\xi}{7R}$.

**Error:** It incorrectly assumes that the branches containing Bulb 2 and Bulb 3 would maintain a series connection after the switch is closed during circuit analysis.

Figure 22: Modeling Flaws on Gemini-2.5-Pro.

# Modeling Flaws

**Question:** 1 mole of helium (ideal gas) undergoes the cyclic process shown in the thermal diagram. In the diagram, *AB*, *BC*, and *CA* are all straight lines, and the relevant parameters have been indicated. Find the cycle efficiency $\eta$.

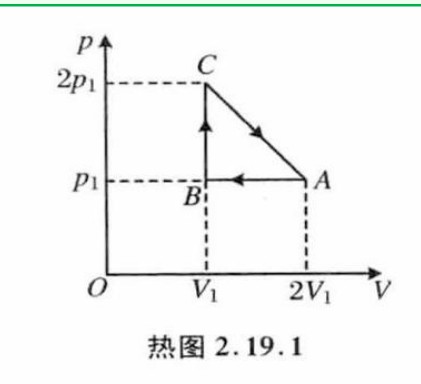

热图 2.19.1

### Response:

...    Process $C \rightarrow A$ (straight line process):   $W_{CA} = 1.5p_1V_1$. Since $T_A = T_C$, the change in internal energy $\Delta U_{CA} = n\,C_V\,(T_A - T_C) = 0$. $Q_{CA} = \Delta U_{CA} + W_{CA} = 1.5p_1V_1$. Since $Q_{CA} > 0$, heat is absorbed.

**Error:** It incorrectly assumes the entire process C→A in the cycle is endothermic during prediction, failing to correctly identify the turning point where the process transitions between endothermic and exothermic phases.

Figure 23: Modeling Flaws on Gemini-2.5-Pro.

# Modeling Flaws

**Question:** The resistance network composed of resistors forms an equilateral triangle as shown in the figure. Given that the resistance of each small segment of the resistor is $r_0$, find the equivalent resistance $R_{BC}$ between points $B$ and $C$.

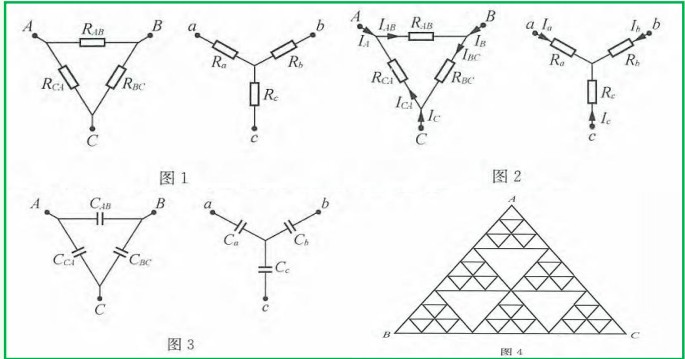

**Response:**

A three-terminal device where the resistance between any pair of terminals (say, 1, 2, 3) is $X$ (i.e., $R_{12} = R_{23} = R_{31} = X$ is equivalent to a Y-network with three identical arms, each of resistance $r$. For this equivalence, $r + r = X$, so $r = X/2$.Thus, each $(k-1)$-th level gasket can be replaced by an equivalent Y-network with arms of resistance $r' = X_{\frac{k-1}{2}}$. Let $O_1$, $O_2$, $O_3$ be the central nodes of the Y-networks corresponding to $G^1_{k-1}$, $G^2_{k-1}$, $G^3_{k-1}$ respectively.

**Error:** It incorrectly applies the Y-$\Delta$ transformation relationship during prediction, leading to errors in both the recursive formula and the final equivalent resistance calculation.

Figure 24: Modeling Flaws on Gemini-2.5-Pro.

## Modeling Flaws

**Question:** A ruby laser emits light with a wavelength of $6943\dot{A}$, which to a very good approximation is a plane wave. What are the amplitudes of the electric and magnetic field vectors of this plane wave propagating through water, if the time-averaged power of the beam in the water is 100 milliwatts/cm^2?

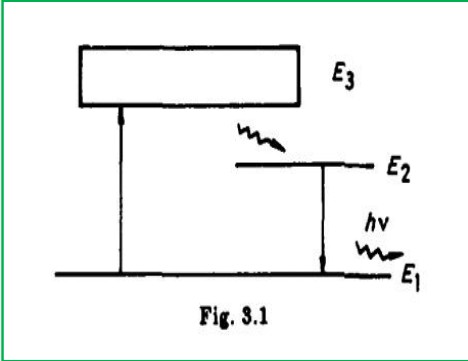

Fig. 3.1

### Response:

The light propagates through water. The refractive index of water is $n \approx 1.33$. The relationship between the intensity $I$ and the amplitude of the electric field $E_0$ in a medium with refractive index $n$ is: $I = \frac{n\epsilon_0 c E_0^2}{2}$, where $\epsilon_0$ is the permittivity of free space ($\epsilon_0 \approx 8.85 \times 10^{-12} F/m$) and $c$ is the speed of light in vacuum. From this, we can find the amplitude of the electric field $E_0 = \sqrt{2I/n\epsilon_0 c}$.

**Error:** In the electric field amplitude calculation procedure, it incorrectly applies the intensity formula for vacuum without properly adjusting for the permittivity in the medium.

Figure 25: Modeling Flaws on Gemini-2.5-Pro.

# Oversimplification

**Question:** A semiconductor block (conductivity σ) has its bottom face (z=0) grounded and top face (z=c) insulated except for a central current-injecting wire. Find the potential φ inside.

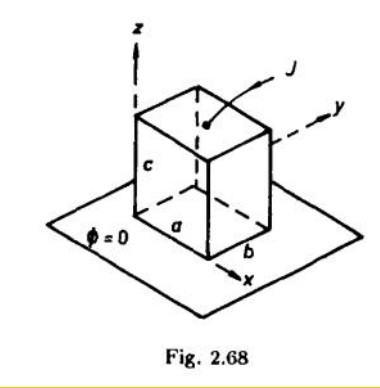

Fig. 2.68

**Response:**

The general solution provided is:
$$\phi(x,y,z) = (A\cos\alpha x + B\sin\alpha x)(C\cos\beta y + D\sin\beta y)\left[E\exp\left(\sqrt{\alpha^2+\beta^2}\,z\right) + F\exp\left(-\sqrt{\alpha^2+\beta^2}\,z\right)\right],$$

**Error:** In presenting the general solution for the potential, the provided formula using separated variables fails to correctly apply the problem's **specific boundary conditions**. It erroneously includes sine function components for the spatial dependencies along x and y, which are incompatible with the insulated side conditions of the block. Additionally, its exponential form for the z-dependency is unadapted, as it neither directly reflects the zero potential requirement at the base of the block nor properly accounts for the distinct linear z-dependence essential for the current-carrying mode that arises when the separation constants for both x and y are zero.

Figure 26: Oversimplification on o4-mini.

# Oversimplification

**Question:** A Three bodies of equal mass *m* and indicated by $i = 1,2,3$ are constrained to perform small oscillations along different coplanar axes forming 120° angles at their common intersection, as shown in Fig. 1.94. Verify that one normal mode is totally symmetric: $x_1(t) = x_2(t) = x_3(t)$, and determine its frequency.

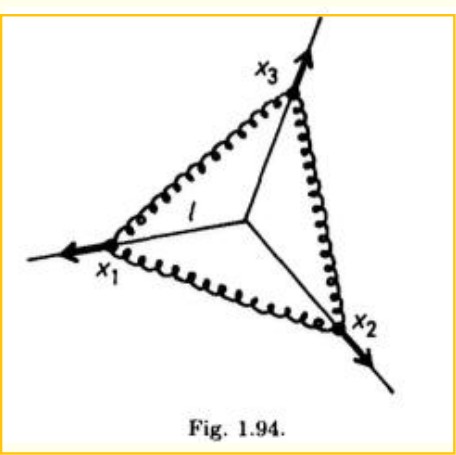

Fig. 1.94.

## Response:

Each pair $(i, j)$ is joined by a spring of spring-constant k and equilibrium length $|R_i - R_j| = \sqrt{3}l$. ... Summing over the three springs gives $V = \sum_{i<j} \frac{3k(x_i + x_j)^2}{8}$.

**Error:** The model builds **incomplete physical model for the potential energy**. It only considers the potential energy from the "coupling springs" (constant k) between the pairs of masses.

Figure 27: Oversimplification on o4-mini.

# Oversimplification

**Question:** In the electrical circuit shown in the figure, $\omega$, $R_1$, $R_2$ and $L$ are fixed; $C$ and $M$ can be varied. Find values of $M$ and $C$ which maximize the power dissipated in resistor $R_2$. What is the maximum power? You may assume, if needed, $R_2 > R_1, \frac{\omega L}{R_2} > 10$.

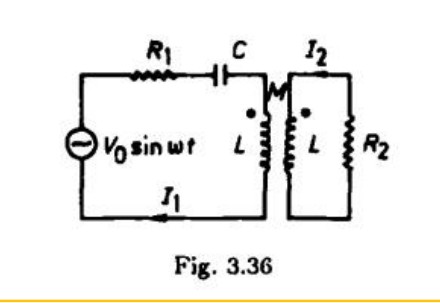

Fig. 3.36

**Response:**

... Secondary: $Z_2 = R_2 + jwL$. ... At resonance in the primary... $C = \frac{1}{w^2 L}$. ... The primary "sees" an extra reflected resistance $R' = (\omega M)^2 / R_2$.

**Error:** The model **oversimplifies of the secondary circuit's** impedance when calculating the reflected impedance. It incorrectly assumes the secondary impedance is purely resistive ($R_2$) and ignores the significant reactance $jwL$ of the secondary inductor.

Figure 28: Oversimplification on o4-mini.

## Oversimplification

**Question:** In An infinite chain of alternating $\pm e$ charges with spacing $a$. Find the potential energy $W_-$ of one negative ion due to all others. *(e = elementary charge, N→∞, Coulomb interaction)*

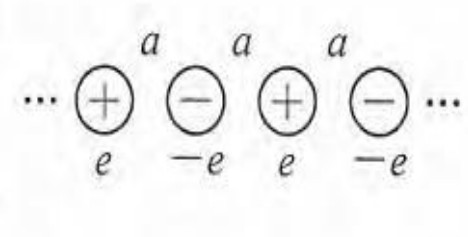

**Response:**

... Other ions are located at distances $n_a$ ($n = 1,2,3\ldots$) from $q_0$...

Summing all $n = 1$ to $\infty$ gives $W_- = \sum_{n=1}^{\infty} U_n$

**Error:** The model only sums for positive n (or in one direction), the calculation omits the interactions with half of the ions in the chain, resulting in a potential energy value that is half of the correct total.

Figure 29: Oversimplification on o4-mini.

# Oversimplification

**Question:** To determine the specific latent heat of fusion of ice, L: an electric heater melts ice at 0∘C. The mass of melted water versus time is recorded at a constant melting rate (see Fig. 3.2). The experiment has two phases: heating at an initial power for 10 minutes, then at an increased power for another 12 minutes. Electrical power is determined from ammeter and voltmeter readings (Fig. 3.3). Calculate L.

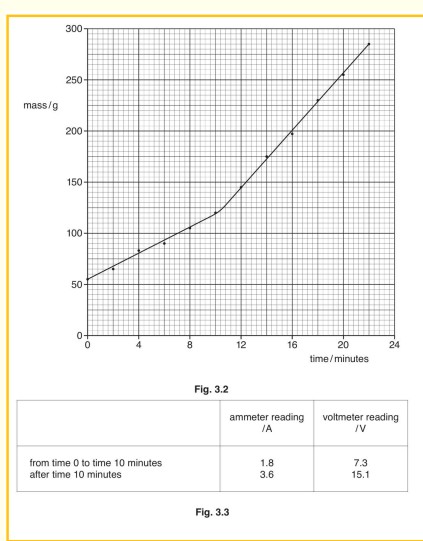

**Fig. 3.2**

|  | ammeter reading /A | voltmeter reading /V |
|---|---|---|
| from time 0 to time 10 minutes | 1.8 | 7.3 |
| after time 10 minutes | 3.6 | 15.1 |

**Fig. 3.3**

## Response:

$... L = \frac{VI}{dm/dt} ... \Rightarrow L_1 = \frac{P_1}{dm/dt} \approx 1.13 \times 10^5 J/kg ... \Rightarrow L_2 = \frac{P_2}{dm/dt} \approx 2.45 \times 10^5 J/kg$ ... The higher-power run is less affected by losses, so we take $L_2 \approx 2.5 \times 10^5 J/kg$.

**Error:** It simply calculates L independently for each heating phase using the formula $L = P/(dm/dt)$ and then selects the value from the higher-power experiment, arguing it's less affected by losses. This method does not properly account for a constant rate of heat exchange (loss or gain) with the surroundings.

Figure 30: Oversimplification on o4-mini.

# Oversimplification

**Question:** A Wollaston prism is made of Iceland spar $n_o = 1.658$, $n_e = 1.486$ with angle $a = 20°$ between the leg and hypotenuse faces of the prism. Determine angle $\beta$ between the linearly polarized rays output from the prism if a ray of natural light is normally incident on the prism.

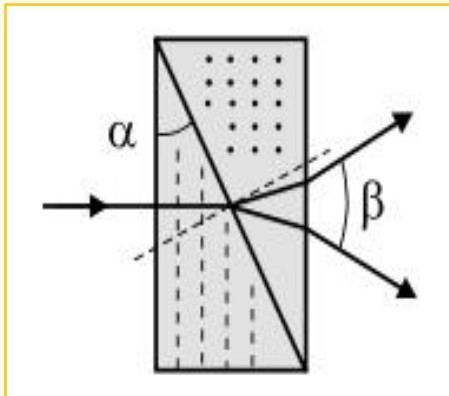

### Response:

In the first prism, the optic axis is parallel to the entrance face and in the plane of the diagram (e.g., vertical). The incident light propagates horizontally. The ordinary ray (o-ray) is polarized perpendicular to the optic axis and the direction of propagation (e.g., out of the plane of the diagram). Its refractive index is $n_o$. The extraordinary ray (e-ray) is polarized parallel to the optic axis (e.g., vertical). Its refractive index is $n_e$. Both rays travel horizontally and are incident on the interface between the two prisms. The angle of incidence on this interface is $a$, measured from the normal to the interface.

**Error:** In the refraction analysis, model fails to correctly apply the angular formula for Wollaston prisms, erroneously considering **only single** refraction without properly calculating the cumulative effect of double refraction.

Figure 31: Oversimplification on Gemini-2.5-Pro.

## False Assumption

**Question:** A The space shuttle $P$ moves in an elliptical orbit ($e = \frac{\sqrt{3}}{2}$) around Earth. At point $D$, it ejects a probe backward, causing the shuttle to enter a circular orbit while the probe escapes on a parabolic trajectory. Find the mass ratio $= \frac{y}{2} = \frac{m1}{m2}$ (shuttle to probe).

### Response:

By momentum conservation： $(m_1 + m_2)v_1 = m_1 v_c + m_2 v_p$(Here, $v_1$ is initial speed, $v_c$ is shuttle's final speed, $v_p$ is probe's final lab speed, all implicitly positive in the same direction.)

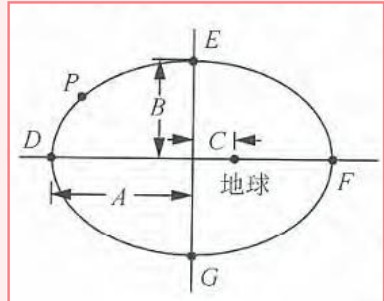

**Error:** It **incorrectly assumes the probe moves forward in the lab frame after being "fired backward."** For the shuttle to speed up significantly (from apogee to circular orbit speed) and the probe to achieve an even higher escape speed, the probe must end up moving backward in the lab frame.

Figure 32: False Assumption on o4-mini.

# False Assumption

**Question:** A particle of mass m is released at $t=0$ in the one-dimensional double square well shown in the figure in such a way that its wave function at $t = 0$ is just one sinusoidal loop (half a sine wave with nodes just at the edges of the left half of the potential as shown). Find the average value of the energy at $t = 0$ (in terms of symbols defined above).

**Response:**

each well of width a sits at **V=0.** Since **V=0 everywhere ψ is nonzero**, the expectation value of the potential is ⟨V⟩=0

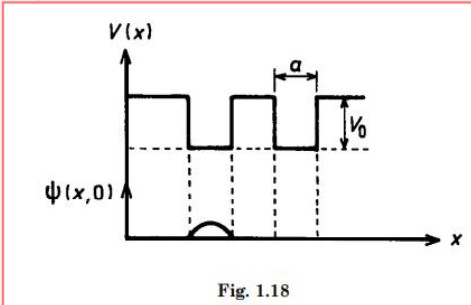

Fig. 1.18

**Error:** The provided figure shows the bottom of the potential wells at a level V0 *below* a dashed line, which typically represents V=0. Thus, inside the well, the potential energy is −V0. However, due to the **incorrect assumption that V(x)=0 inside the well**, it erroneously concludes that the expectation value of the potential energy ⟨V⟩=0.

Figure 33: False Assumption on o4-mini.

# False Assumption

**Question:** The figure shows a schematic diagram of a large steam pile driver. The iron tower is 40 m high, and the hammer has a mass of 10 t. A reinforced concrete pile with a length of 38.5 m is now being driven into the ground. The mass of the pile is 24 t, and its cross-section is a square with an area of **0.25 m²**. The resistance per unit area on the side of the pile due to soil is **k = 2.65 × 10⁴   N/m²**.

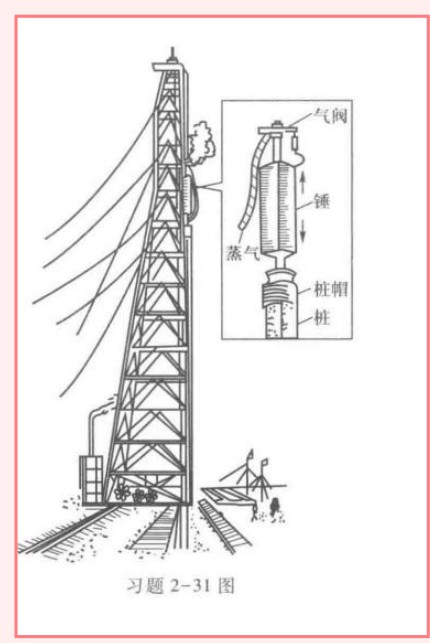

习题 2–31 图

**Response:**

The pile sinks until its weight is balanced by the total frictional force, i.e., $W = F_f$. $235200 = (5.30 \times 10^4) \times h$. $h \approx 4.437735849m$.

**Error:** It incorrectly treats the resistance as a constant force and directly applies force equilibrium in the prediction, neglecting the fact that the resistance increases linearly with depth.

Figure 34: False Assumption on Gemini-2.5-Pro.

