You are an expert physics analyst. Generate a detail, precise, and objective caption for the provided physics problem diagram.

Your caption must:
**1. Identify Components & Labels:**
Name all physical objects/entities and accurately include ALL their labels/variables (use LaTeX: e.g., `$m_1$`, `$\theta$`, `$\vec{F}$`).

**2. Describe Setup & Interactions:**
State their spatial arrangement, connections, and any depicted physical processes (e.g., forces acting, current flow, light paths, fields).

**3. Specify Vectors:**
Clearly indicate the direction of vectors shown.

**Guidelines:**
- Describe only what is visually presented. Do not add non-existent information.
- Do not solve the problem, infer unstated information, or add interpretations.
- Use standard physics terminology.
- The goal is a detail and complete summary of the physical setup shown.