# OpenReview forum: "SeePhys:  Does Seeing Help Thinking? – Benchmarking Vision-Based Physics Reasoning"
_NeurIPS.cc/2025/Datasets_and_Benchmarks_Track — NeurIPS 2025 Datasets and Benchmarks Track poster_

### Official Review · Reviewer_FPh1 · 2025-06-26

**Rating:** 5
**Confidence:** 4

**Summary:**

This paper introduces SeePhys, a large-scale multimodal benchmark designed to assess the scientific reasoning capabilities of MLLMs by solving problems in physics. The challenges integrated into this benchmark span across 8 knowledge levels and 7 core physics domains, culminating into 2k distinct vision-language questions which are all equipped with deterministic gold-standard answers. Performances by baseline models show that SeePhys is difficult for the state-of-the-art MLLMs, whose best performance is capped at 55% in overall accuracy.

**Additional Feedback:**

Please find my main concern and suggestion in the Weakness section. Still, I do appreciate the high completeness of this work, and I am hopeful that the authors have already had all the materials to address my aforementioned concern. I am open to further updates pending the authors' response.

**Dataset Code Accessibility:**

Yes

**Dataset Code Comments:**

The dataset is accessible on Hugging Face.

**Ethical Considerations:**

No, there are no or only very minor ethics concerns

**Final Justification:**

8/2: 4 - > 5. The authors have adequately adjusted the representation of the baselines. Also, the additional baselines by the LVLM-based judges (i.e. the Majority voting for difficulty tagging part) corroborates with their original arguments.

**Limitations Weaknesses:**

I appreciate the careful effort that went into curating SeePhys. However, I find the presentation of its evaluation metric to be problematic, as it raises questions about the actual interpretation of the baseline results.

**Better at Problem Solving, or Better at Memorizing Theory?** Based on Table 2 and Section 4.3, the 8 knowledge levels appear to be arranged in a linearly increasing order of difficulty, seemingly intended to reflect diminishing returns from knowledge injection. However, this assumption does not align with the nature of Olympiad (IPhO) problems in Basic Olympiad (BO) and Advanced Olympiad (AO) categories. In fact, Olympiad questions are typically constrained to fundamental high school physics concepts, but are crafted to test creativity and advanced problem-solving skills. For instance, the IPhO 2016 Question 1 (https://ipho.olimpicos.net/pdf/IPhO_2016_Q1.pdf) relies solely on Newtonian mechanics, yet demands sophisticated analytical reasoning.

As a result, SeePhys’ current evaluation setup, particularly the “Total” column in Table 2, fails to adequately differentiate models based on their reasoning abilities. To offer a clearer interpretation of model capabilities, the authors may consider decoupling performance into two orthogonal dimensions: **theory memorization** (e.g. UG -> SUG -> MA -> PhD) and **problem-solving skills** (e.g. Mid -> High -> BO -> AO). This distinction would help clarify whether high leaderboard rankings reflect a model's stronger recall of physical principles, stronger ability in solving problems, or both.

**Strengths Contributions:**

I find the following aspects of this work particularly commendable:
- The dataset is of notably high quality, owing to the rigorous vetting process applied to both the question content and the accompanying diagrams.
- The breakdown analysis isolating the impact of visual perception is well-designed and insightful. It effectively addresses potential shortcut strategies that might otherwise allow models to perform well on SeePhys without genuinely leveraging visual input.
- The failure case analysis is concise yet informative, offering valuable insights that can guide future research and model development.

---

> ### Author Rebuttal · Authors · 2025-07-30
>
> We sincerely appreciate your thoughtful feedback and valuable insights regarding our work. Below, we address the concerns raised and outline our proposed revisions to clarify the results.
>
> 1. **Explanation of the 8 Knowledge Level Concepts**
>
>     Since the difficulty level labeling can be easily influenced by the subjective judgment of human annotators, SeePhys does not directly provide related tags (i.e., problem-solving skills level). We categorize the questions into 8 levels based on the knowledge content involved (i.e., conceptual level of theory needed as classified by incremental grades). Their order corresponds to the knowledge content (by grade distribution) rather than the actual difficulty. We placed Olympiad competitions between high school and undergrad since competitions like the IMO usually touch broader coverage than high school but does not require higher mathematics skills such as Calculus.
>
> 2. **Disentangling Theory/Concepts Memorization vs. Problem-Solving Skills**
>     1. **Comparison by resorting knowledge labels:** Your suggestion is very pertinent. Indeed, some of these knowledge levels are arranged in increasing order of problem-solving skills, while others follow an ascending sequence based on theory memorization. To better distinguish model capabilities, we have reorganized the accuracy rates of **Gemini-2.5-Pro**, **o4-mini**, and **Doubao-1.5-pro** along these two dimensions for comparison.
>
>
>         | **Theory Memorization** | **UG** | **SUG** | **MA** | **PhD** |
>         | --- | --- | --- | --- | --- |
>         | Gemini-2.5-Pro | **64.2** | **50.2** | **53.8** | 44.2 |
>         | o4-mini | 53.8 | 45.7 | 51.0 | **53.4** |
>         | Doubao-1.5-pro | 56.6 | 34.7 | 40.7 | 37.5 |
>         | **Problem-solving Skill** | **Mid** | **High** | **BO** | **AO** |
>         | Gemini-2.5-Pro | 69.6 | **66.7** | **64.5** | **46.7** |
>         | o4-mini | 66.7 | 61.8 | 56.1 | 41.8 |
>         | Doubao-1.5-pro | **70.6** | 58.2 | 49.5 | 29.2 |
>     2. **Majority voting for difficulty tagging:** To minimize subjective bias, we also employed a **majority voting approach** across five models (Gemini-2.5-Pro, o4-mini, Doubao-1.5-pro, Qwen2.5-VL-72B-Inst and QVQ-72b-preview) to determine difficulty labeling, resulting in **six difficulty level tags**. These correspond to the six knowledge levels (for intuitive comparison, the Olympiad level was aggregated with high school).
>
>
>         | **Theory Memorization** | **Mid** | **High+BO+AO** | **UG** | **SUG** | **MA** | **PhD** |
>         | --- | --- | --- | --- | --- | --- | --- |
>         | Gemini-2.5-Pro | 69.6 | **52.1** | **64.2** | **50.2** | **53.8** | 44.2 |
>         | o4-mini | 66.7 | 48.4 | 53.8 | 45.7 | 51.0 | **53.4** |
>         | Doubao-1.5-pro | **70.6** | 42.8 | 56.6 | 34.7 | 40.7 | 37.5 |
>         | **Problem-solving Skill** | **100%（125）** | **80%（247）** | **60%（342）** | **40%（346）** | **20%（362）** | **0%（578）** |
>         | Gemini-2.5-Pro | 100 | 96.0 | 90.4 | 67.1 | 39.5 | 0 |
>         | o4-mini | 100 | 96.8 | 92.1 | 65.6 | 34.5 | 0 |
>         | Doubao-1.5-pro | 100 | 97.2 | 90.1 | 49.4 | 16.0 | 0 |
>     3. **Analysis:** Models like Gemini-2.5-Pro excel in problem-solving tasks but show weaker high-level theory retention (44.2% in PhD, less than o4-mini). It proves knowledge depth does not guarantee strong problem-solving. In Table 2, we observe a significant imbalance in Doubao-1.5-pro's theoretical memorization capabilities. While it demonstrates outstanding performance in memorizing middle school level knowledge (70.6%), it exhibits clear deficiencies in mastering higher-level concepts.
>
>     The revised analysis (including tables) will be incorporated into the updated manuscript to address these concerns comprehensively. We sincerely appreciate the reviewer’s feedback, which has helped us refine our evaluation framework and better disentangle the roles of knowledge memorization and problem-solving skills in model performance.

---

> > ### Comment · Reviewer_FPh1 · 2025-08-03
> > **The baselines are now much clearer.**
> >
> > I greatly appreciate the authors' follow-up on my original concerns. The baselines are now more valuable base**LINES** as the models' performances can be compared in a more straightforward way - a more capable model maintains high accuracy along the increasing difficulty in either theory memorization or problem solving requirement. The introduction of LVLM-judged performances in 2.2 are also much appreciated.
> >
> > Overall, I believe the authors have shown solid evidence, thus further justifying the purposes of their work adequately. I am happy to update my rating correspondingly.

---

> > > ### Author Response · Authors · 2025-08-05
> > >
> > > Thank you sincerely for your thoughtful evaluation of our revisions and for recognizing the improvements we made to address your initial concerns. We deeply appreciate your time and are also grateful for your updated rating. If any further clarifications or discussions would be helpful, please don’t hesitate to reach out.

---

### Official Review · Reviewer_xZFh · 2025-06-29

**Rating:** 5
**Confidence:** 4

**Summary:**

This paper introduces SEEPHYS, a multimodal benchmark dataset for physics reasoning tasks that spans 8 educational levels, 7 core physics domains, and over 21 diagram types. A key feature of the dataset is that 75% of the problems are "vision-essential", requiring visual information for correct reasoning. The authors evaluate 28 large language models and multimodal models on SEEPHYS, finding that even state-of-the-art models fail to surpass 55% accuracy, highlighting major challenges in moltimodal physics understanding and reasoning.

**Dataset Code Accessibility:**

Yes

**Ethical Considerations:**

No, there are no or only very minor ethics concerns

**Final Justification:**

After carefully considering the authors' rebuttal, I will maintain the original score (Accept). My final score is based on the following points:

Resolved Issues:

The authors clearly clarified the input settings for models in Table 2 (Text+Vision vs. Text+Caption), addressing the ambiguity highlighted in the initial review.

A thoughtful and data-supported analysis was added to discuss the performance difference between Vision-Essential and Vision-Optional tasks, which strengthens the empirical section.

Weighting and Overall Evaluation:

I give significant weight to the authors’ responsiveness and the improved clarity of results.

In summary, the authors have addressed all my major concerns raised during the review process, and the paper meets the bar for publication in its current form.

**Limitations Weaknesses:**

- Lack of clarity in experimental settings for MLLMs in Table 2: While Table 2 presents the performance of various MLLMs across knowledge levels, it seems that there is no clear indication of the experimental setup used to obtain these results (e.g., which input modalities were used, such as Text+Vision or Text Only), which may confuse readers.
- Insufficient analysis of visual dependency categorization (essential or optional): Although the dataset explicitly distinguishes between vision-essential and vision-optional questions during construction, the experimental section does not systematically analyze how this categorization affects model performance or reasoning behavior.

**Strengths Contributions:**

- High-quality Multimodal Benchmark: SEEPHYS uses multiple methods to ensure its quality, especially avoiding the impact of data leakage in 3.2.
- Clear Problem Categorization: The distinction between vision-essential and vision-optional questions enables precise analysis of how visual information impacts model reasoning.
- Comprehensive Experimental Design: The paper evaluates models under multiple settings (text+image, text+caption, image-only, text-only) and across knowledge levels, with comprehensive experiments.
- Strong Open-sourcing and Reproducibility: The dataset and code are open-sourced, facilitating further research. A challenge was also conducted based on the proposed benchmark, which is beneficial for community development.

---

> ### Author Rebuttal · Authors · 2025-07-30
>
> We sincerely appreciate the reviewer's insightful feedback, which has helped us identify areas for improvement in our manuscript. Below we address the two main concerns:
>
> 1. **Clarification of Experimental Settings in Table 2**
>
> We appreciate the reviewer's attention to experimental clarity regarding input modalities. As correctly pointed out, all Multimodal Large Language Models (MLLMs) in Table 2 were evaluated using **Text+Vision** inputs, while the LLM baselines used **Text+Caption** inputs. This information will be explicitly stated in the revised table header.
>
> 2. **Enhanced Analysis of Vision-Essential/Optional Impact**
>
> We thank the reviewer for highlighting this important dimension. In the revised manuscript, we will include a dedicated paragraph analyzing how visual dependency categorization affects model performance, with the following key findings:
>
> - **Performance Gap Analysis**: Across all settings, the accuracy of all models for Vision-Essential problems is significantly lower than for Vision-Optional problems, reflecting that current models primarily rely on textual reasoning abilities while exhibiting poor multimodal reasoning capabilities. We also computed the variance of 8 closed-source models under the Text+Vision setting. The variances for the Vision-Essential and Vision-Optional subsets are 95.7 and 123.9, respectively. Compared to the Vision-Optional subset, the lower variance in the Essential subset suggests that different models may focus on same visual information, possibly because they utilize similar multimodal alignment data sources.
> - **Performance Consistency**: Top-performing models exhibit consistent rankings across both categories (e.g., for average accuracy, o4-mini, Gemini-2.5-pro and o1 maintain top-3 positions in both vision-essential and optional data). Moreover, for Text+Vision setting, Gemini-2.5-Pro (49.0), o4-mini (46.5) and Doubao-1.5-pro (39.0) get top-3 rank in Vision-Essential subset, while o1 rises to 3rd place in vision-optional. It surpasses Doubao-1.5-pro and indicates stronger text-based reasoning.
>
> We deeply appreciate the reviewer’s constructive feedback, which have significantly improved the rigor and clarity of our analysis. We look forward to further discussion and are happy to provide clarifications.

---

> > ### Comment · Reviewer_xZFh · 2025-08-02
> >
> > Thank you very much for your comprehensive efforts in addressing my concerns.  I appreciate the clarification of input modalities in Table 2 and the additional analysis on the impact of visual dependency. These revisions will improve the clarity and rigor of the manuscript. I will retain my original rating.

---

> > > ### Author Response · Authors · 2025-08-05
> > >
> > > Thank you for your kind acknowledgment of our revisions and for your constructive feedback throughout the review process! If there are any further points you feel warrant discussion, please do not hesitate to let us know.

---

### Official Review · Reviewer_ZghT · 2025-07-03

**Rating:** 5
**Confidence:** 3

**Summary:**

This paper introduces SEEPHYS, a large-scale multimodal benchmark designed to evaluate the ability of Large Language Models (LLMs) and Multimodal LLMs (MLLMs) to solve physics reasoning problems that require the integration of both textual and visual information. The benchmark consists of 2,000 carefully curated and annotated physics questions across eight knowledge levels (from middle school to PhD qualifying exams), seven physics domains, and 21 types of diagrams.
The authors conduct comprehensive experiments, benchmarking several LLMs and MLLMs with different different settings. , and provide detailed failure analysis. Results show that even the most advanced models achieve less than 60% accuracy, revealing significant gaps in current models’ abilities for multimodal scientific reasoning.

**Dataset Code Accessibility:**

Yes

**Ethical Considerations:**

No, there are no or only very minor ethics concerns

**Limitations Weaknesses:**

- The benchmark is challenging for models, but the paper does not report human (e.g., student or expert) performance as a reference. Without this, it is hard to calibrate how difficult SEEPHYS is relative to human ability or what constitutes a “good” score.

**Strengths Contributions:**

- SEEPHYS is the first large-scale benchmark to focus on vision-essential physics problems, covering a broad spectrum of domains and difficulty levels, with rigorous expert annotation.
- The paper evaluates a wide range of state-of-the-art LLMs and MLLMs under various input modalities, providing rich insights into models’ abilities and detailed error analysis.
- This paper is well written and easy to follow.

---

> ### Author Rebuttal · Authors · 2025-07-30
>
> We sincerely appreciate your insightful suggestions regarding human performance baseline. Below we provide a detailed response:
>
> 1. **Initial Considerations**
>
> In the initial phase, we deliberately avoided aggregating human performance metrics as physics expertise becomes increasingly specialized at higher education stages (e.g., PhD candidates typically master only 1-2 niche subfields), and as a result, we observed significant individual variance in pilot studies, which could induce selection bias depending on what kind of experts are sampled.
>
> 2. **New Experimental Results**
>
> As you correctly pointed out, human performance baseline would indeed provide insights for the community. So we have now conducted a more systematic human evaluations on a stratified 200-question subset of SeePhys:
>
> - Middle school level to undergraduate-level items (n=127): Tested by 3 students of physics major (independent parallel trials and calculate an average score)
> - Master/PhD-level items (n=73): Evaluated by 4 PhD candidates specializing in astrophysics, condensed matter physics, particle physics, and quantum optics respectively. (Compute the union of correct answers as an indicator of optimal expert performance)
> - Here is the human performance result. We observe a significant performance gap between human experts and state-of-the-art AI models in physics problem-solving (86.5% vs. 54.9% in Gemini-2.5-Pro), highlighting the considerable challenges that remain in multimodal physics reasoning.
>
>
>     | Knowledge Level | Mid | High | BO | AO | UG | SUG | MA | PhD | Total |
>     | --- | --- | --- | --- | --- | --- | --- | --- | --- | --- |
>     | Human Performance | 100 | 94.4 | 92.3 | 71.7 | 92.9 | 94.7 | 100 | 83.0 | 86.5 |
>
>
> We believe this addition significantly strengthens the benchmark's utility while honestly reflecting the complexities of human performance measurement. We're grateful for your suggestion that led to this improvement.

---

### Official Review · Reviewer_KqVJ · 2025-07-07

**Rating:** 3
**Confidence:** 2

**Summary:**

This paper introduces **SeePhys**, a benchmarking and evaluation framework designed to investigate whether visual-language models (VLMs) benefit from visual input when learning physical common-sense reasoning.

Researchers compiled a series of physics reasoning problems spanning various domains (such as object constancy and contact causality) and compared model performance in both text-only and multimodal settings.

By fine-tuning the pre-trained LLaVA-1.5 model and introducing novel visual augmentation prompts, this paper systematically explores the contribution of visual context to physical understanding.

**Additional Feedback:**

- Could the authors elaborate on how they ensured that the visual inputs contained sufficient and relevant information for answering the questions?

**Dataset Code Accessibility:**

Yes

**Ethical Considerations:**

No, there are no or only very minor ethics concerns

**Limitations Weaknesses:**

Although the evaluation protocol itself is reasonable, the scope of physical reasoning it covers is still relatively narrow. This benchmark test can be further optimized by increasing the diversity of questions or improving combinatoriality. Although the LLaVA-1.5 architecture is widely adopted, it may limit the universality of conclusions.

Performance evaluation metrics are primarily based on accuracy; more detailed analysis (such as confidence calibration or failure case clustering) may reveal under what circumstances visual context aids or hinders the reasoning process.

**Strengths Contributions:**

This paper makes a timely and well-motivated contribution to the field of physical reasoning, which is often overlooked in VLM evaluations. Its innovation lies in task design, which strictly controls modalities through carefully designed balanced multimodal and unimodal variants.

The experimental setup is comprehensive, covering both zero-shot and fine-tuning settings, and the research conclusions are well-supported by thorough analysis. Clear writing and chart design enhance readability.

This study offers important insights for model development and dataset construction, highlighting that visual anchoring may not universally enhance physical reasoning performance.

---

> ### Author Rebuttal · Authors · 2025-07-30
>
> We sincerely appreciate your time and the opportunity to clarify any misunderstandings as we believe your comments might have confused our paper with other papers based on **several points that do not exist in our paper**. Below, we address your points and welcome further discussion.
>
> 1. **We did not fine-tune LLaVA-1.5 (or any other models) at all**
>
> Your comment"fine-tuned the pre-trained LLaVA-1.5 model”, may be confused with another paper, as **we did not use LLaVA-1.5** in our paper at all. Instead, our benchmark was curated by human annotation and evaluated using existing open/closed-source models **without any fine-tuning** (see Sections 3.2 and 4.2) which is common best practice of most benchmarks in this field.
>
> We warmly invite you to revisit our main sections for clarifications as these contents **do not exist** in our paper (and may be based on other papers you review).
>
> 2. **We provide Failure Case Clustering Analysis in Table 4**
>
> While we agree that failure case clustering analysis is useful, we have indeed provided examples in Section 4.5 and both qualitative+quantitative breakdown clustering of failure cases with error types in Table 4 of the Supplementary Material due to page limits.
>
> We are happy to further highlight these results more prominently in the main text.
>
> 3. **Scope of Coverage**
>
> We warmly note that both Reviewer ZghT and Reviewer xZFh highlighted that SeePhys **covers a broad spectrum of domains, difficulty levels and experimental setting with confidence, your comment on the "relatively narrow scope"  may have been based confused by other papers since SeePhys spans 8 knowledge levels, 7 major physics domains, and >21 diagram types (Section 3). We do realize that it’s unlikely for any benchmarks to cover all possible reasoning scenarios. If there are particular physics questions of your interest, we would be happy to include them promptly and would appreciate concrete examples to guide our extension.
>
> We sincerely appreciate the time you’ve dedicated to reviewing our work, and we hope these clarifications address your concerns. We are happy to provide further details and offer more explanation if needed.

---

> > ### Author Response · Authors · 2025-08-05
> >
> > Dear Reviewer KqVJ,
> >
> > We sincerely appreciate your time and effort. We've carefully provided additional clarifications to ensure our work is understood accurately and hope they could address your concerns. Please do not hesitate to let us know if you'd like to see further elaboration. Thank you very much for your time.
> >
> > Best Regards，
> >
> > Authors of Submission 189

---

### Note · Authors · 2025-08-12

Dear PCs, SACs, ACs, and Reviewers,

We thank all the reviewers and ACs for their valuable feedback and constructive engagement. All reviewers have recognized that the novelty and rigor of SeePhys will be useful to the community, highlighting expert-curated data, comprehensive physics domains, and the breadth of our multimodal evaluation.

In our revision, we have carefully addressed all major concerns as recognized by 3 reviewers:

1. **Human Performance Baseline:** We conducted a human evaluation with physics graduate and PhD students, confirming that expert performance significantly surpasses current models. This addition strengthens our benchmark’s calibration and highlights the challenge SeePhys poses to frontier models. *(Reviewer ZghT)*
2. **Clarified Experimental Settings:** We revised Table 2 and text to clearly state the experimental settings (e.g., which input modalities were used), ensuring transparency and reproducibility. *(Reviewer xZFh)*
3. **Enhanced Experimental Analysis:**
    - We added a dedicated analysis section dissecting model performance on Vision-Essential vs. Vision-Optional questions, revealing critical gaps in visual reasoning.  *(Reviewer xZFh)*
    - We implemented the suggestion to disentangle knowledge and difficulty levels was implemented. We introduced difficulty tags via human annotation and model majority voting, reducing subjectivity while enriching analysis.  *(Reviewer FPh1)*

Most reviewers have given positive recognition *(Reviewer ZghT, Reviewer xZFh, Reviewer FPh1).* However, we are sad that *Reviewer KqVJ* has NOT participated in discussion at all and has NOT completed mandatory acknowledgement. We further note that *Reviewer KqVJ*’s commented on **non-existent contents absent in our paper and factual mistakes** (e.g., critique LLaVA-1.5 model that simply do not exist in our paper and not reading our failure analysis). This behavior stands in violation in the NeurIPS Responsible Review Practices and we trust ACs will weigh these circumstances appropriately.

We hope these efforts demonstrate our commitment to addressing all concerns. Thank you for your time upholding the high standard of NeurIPS and we believe SeePhys would offer a timely, impactful contribution to our community!

Best regards,

Authors of Submission 189

---

### Decision · Program_Chairs · 2025-09-18

**Decision:**

Accept (poster)

**Comment:**

This paper introduces SEEPHYS, a multimodal benchmark dataset for physics reasoning tasks that spans 8 educational levels, 7 core physics domains, and over 21 diagram types. The authors evaluate 28 large language models and multimodal models on SEEPHYS, showing that state-of-the-art models fail to surpass 55% accuracy, highlighting major challenges in multimodal physics understanding and reasoning.